

# Stress correlations in near-crystalline packings

Roshan Maharana$^\star$ and Kabir Ramola$^\dagger$

Tata Institute of Fundamental Research, Hyderabad 500107, India

$\star$ roshanm@tifrh.res.in , $\dagger$ kramola@tifrh.res.in

## Abstract

We derive exact results for stress correlations in near-crystalline systems in two and three dimensions. We study energy minimized configurations of particles interacting through Harmonic as well as Lennard-Jones potentials, for varying degrees of microscopic disorder and quenched forces on grains. Our findings demonstrate that the macroscopic elastic properties of such near-crystalline packings remain unchanged within a certain disorder threshold, yet they can be influenced by various factors, including packing density, pressure, and the strength of inter-particle interactions. We show that the stress correlations in such systems display anisotropic behavior at large lengthscales and are significantly influenced by the pre-stress of the system. The anisotropic nature of these correlations remains unaffected as we increase the strength of the disorder. Additionally, we derive the large lengthscale behavior for the change in the local stress components that shows a $1/r^d$ radial decay for the case of particle size disorder and a $1/r^{d-1}$ behavior for quenched forces introduced into a crystalline network. Finally, we verify our theoretical results numerically using energy-minimised static particle configurations.



---

# 1 Introduction

Jammed athermal materials find relevance in various fields such as soft condensed matter physics, material science, civil engineering and metallurgy [1, 2]. Additionally, jammed packings also arise in fields such as biophysics, where cellular tissues are well described by soft potential models [3, 4]. They arise when it is not feasible to achieve true thermodynamic equilibrium. The stability of athermal solids against mechanical disturbances can be attributed to the macroscopic rigidity arising from the network of constituent particles [5–9]. This collective elasticity arises in any system of interacting particles at low temperatures and is observed universally in both crystalline and amorphous structures of athermal solids [10–24]. The reference states that make up the collection of amorphous solids are highly dependent on the preparation method, and each configuration satisfies the conditions of local equilibrium i.e., the force and torque balance of each constituent. Amorphous structures, while stable in a local sense, are typically not the lowest energy states of their individual components [25–27]. Consequently, jammed packings of soft particles can exhibit both amorphous as well as crystalline structures.

Near-crystalline materials demonstrate a range of unique properties and serve as a bridge between the physics of crystals and amorphous materials, providing valuable insights into the behavior of athermal ensembles [28–36]. The large-scale elasticity properties exhibited in both amorphous and crystalline solids link these two typically distinct branches of condensed matter physics [30, 37, 38]. Recent studies on near-crystalline materials have revealed various characteristics similar to those found in fully amorphous materials, including the presence of quasi-localized modes [39–41]. Such near-crystalline materials therefore help in establishing a connection between the well studied physics of crystals and that of amorphous solids by introducing disorder gradually into athermal crystalline packings. On the one hand, while

crystals have ordered structures, amorphous solids are characterized by random and inflexible structures that arise from the competing interactions between constituent particles. Despite their distinct local structures [42], crystalline and amorphous packings exhibit many similar elastic properties [18,43]. Given the long-range displacement correlations in these systems [44,45], one may reasonably question whether the microscopic structure affects the large-scale elasticity properties [46]. Therefore a crucial question to address is how global rigidity is manifested in distinct networks and whether this can be detected in the local stress tensor fluctuations [47,48]. Although some properties of athermal ensembles can be described by temperature-like variables [49,50], and despite several attempts at a unifying framework, there is still a lack of understanding of these properties in non-isotropic materials and near-crystalline systems. It is therefore of interest to generate ensembles that can be precisely characterized theoretically. In this paper, we develop a microscopic theory for stress correlations in near-crystalline systems arising from various types of disorders such as particle size disorder or due to quenched forces.

Stress correlations are a key ingredient in understanding the physics of disordered systems, and there have been several recent studies that establish their importance in amorphous materials [51,52]. Such correlations provide valuable insights into the collective behavior of interacting particles and are widely used in fields such as material science, fluid dynamics, and geophysics [19,53–55]. Understanding stress correlations can help us predict the strength and stability of materials under various external conditions [56,57] as well as give insights into the rheology of particulate packings, such as their ability to flow or resist deformation. Stress correlations provide a deeper insight into the degree of rigidity or floppiness within particle packings and how they react to external influences like shear or compression [17,43]. Recently, stress correlations have also been studied in various types of systems such as glasses, granular packings, and gels amongst others, and has become a question that has attracted considerable interest [17–23]. There have been several theoretical studies that use material isotropy and homogeneity in amorphous materials to derive the large length scale anisotropic behavior of the stress correlations [58–60]. Several numerical studies have also explored stress correlations in computer-simulated disordered packings [22,23,61].

The main results of this paper can be summarized as follows. We derive the displacement fields due to the introduction of particle size disorder or due to external quenched force in a crystalline system through a microscopic disorder perturbation expansion. Using the linear order displacement and force fields, we derive the components of the change in the local stress tensor on each grain. At large lengthscales, the local stresses show anisotropic $1/r^d$ radial behavior for particle size disorder and $1/r^{d-1}$ behavior for external force quenching. We analyze the local pressure fluctuation which shows similar radial behavior yet isotropic at large lengthscales. We also measure the global bulk and shear modulus for such a near-crystalline system which show excellent match with simulations for finite small disorder. We then derive the configurational averaged correlations of the local stress fluctuations which are verified through numerical simulations in two different models in both two and three dimensions. We show that the stress correlations in disordered crystals show different behavior to that of an isotropic amorphous material at a high packing fraction or high pressure limit.

The outline of the paper is as follows. In Section 2 we introduce the microscopic models, while in Section 3, we present the corresponding preparation protocols. In Section 4 we employ the microscopic approach to derive macroscopic properties, specifically the bulk and shear moduli of near-crystalline systems. In Section 5 we present a detailed derivation of the displacement fields due to the introduction of microscopic disorder in the crystalline packing. In section 6 we derive the change in the local stress tensor components and their correlations through this microscopic approach and compare these predictions against direct numerical simulations. Additionally, we draw parallels between these results and those obtained in the

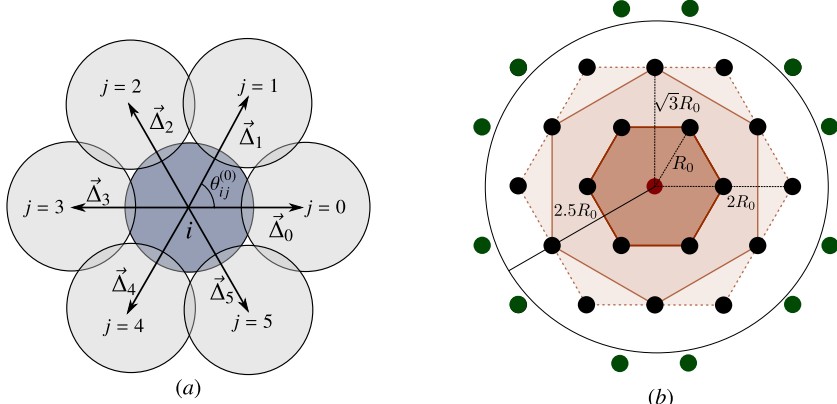

Figure 1: Schematic of particle arrangement in a hexagonal close packing for two different models considered in this paper. (a) Soft repulsive interactions with a Harmonic potential, where the neighboring particles overlap with the central particle. (b) Lennard-Jones (LJ) interaction with a cutoff. Here, the outermost circle corresponds to the interaction range with respect to the central particle (red).

recently developed VCTG framework for amorphous systems. In Section 6.5, we extend our theory to three dimensional fcc arrangement of particles. In Section 7, we derive the displacement and force fields due to point forces. In Section 8, we derive the distributions of change in local stresses from the linear relations between the stress and microscopic disorder in a near-crystalline system. Finally, we conclude and provide directions for future investigations in section 9.

## 2 Models

We study two well-known canonical glass-forming model potentials that can be used to create amorphous, as well as near-crystalline structures: short-ranged Harmonic interactions, and an attractive Lennard Jones interaction with a cutoff. A schematic of the particle neighbourhoods for both models are shown in Fig. 1.

### 2.1 Short-ranged repulsive harmonic interaction

For the case of the short-ranged Harmonic model, we examine systems consisting of frictionless soft disks in two dimensions and spheres in three dimensions with different levels of overcompression. These particles interact with one another through a one-sided pairwise potential [14, 31, 62] which takes the following form:

$$V_{a_{ij}}\left(\vec{r}_{ij}\right) = \frac{K}{\alpha}\left(1 - \frac{|\vec{r}_{ij}|}{a_{ij}}\right)^{\alpha}\Theta\left(1 - \frac{|\vec{r}_{ij}|}{a_{ij}}\right),\tag{1}$$

where $\vec{r}_{ij}$ represents the displacement vector between particles $i$ and $j$, situated at positions $\vec{r}_i$ and $\vec{r}_j$, respectively. Here $a_{ij}$ are called the quenched interaction lengths that are defined as the sum of individual radii, denoted as $a_{ij} = a_i + a_j$. In this study, we select $\alpha = 2$ to establish a harmonic pairwise potential between the particles. The length parameters $a_{ij}$ are then set as follows [32, 35, 36, 44, 45, 63]

$$a_{ij} = 2a_0 + \eta a_0(\zeta_i + \zeta_j),\tag{2}$$

where $a_0$ is the radius of each particle in the crystalline state. The variables $\zeta_i$ are independent and identically distributed random numbers drawn from a uniform distribution ranging between $-1/2$ and $1/2$, that are individually assigned to each particle within the system. The parameter $\eta$ (polydispersity) controls the magnitude of the disorder.

## 2.2 Attractive Lennard-Jones interaction with cut-off

We investigate particles interacting via long-ranged power-law potentials, which are smoothened up to the second order at a specified cutoff interaction length ($r_{ij}^c$), set at $2.5a_{ij}$ [29,41]. This cut-off is set to speed up the numerical simulations and for most purposes, a cut-off greater than 1.5 yields similar mechanical properties [64]. The smoothened LJ potential for a cut-off interaction length $2.5a_{ij}$ can be represented as

$$V_{a_{ij}}(\vec{r}_{ij}) = 4K \left[ \left( \frac{a_{ij}}{|\vec{r}_{ij}|} \right)^{12} - \left( \frac{a_{ij}}{|\vec{r}_{ij}|} \right)^6 + \sum_{l=0}^{2} c_{2l} \left( \frac{|\vec{r}_{ij}|}{a_{ij}} \right)^{2l} \right] \Theta \left( 2.5 - \frac{|\vec{r}_{ij}|}{a_{ij}} \right), \qquad (3)$$

The disorder is introduced through the length parameters $a_{ij}$ and can be represented as [29, 41],

$$a_{ij} = \begin{cases} \lambda_{SS}, & \text{both } i, \text{and } j \text{ are unlabeled,} \\ \eta(\lambda_{SL} - \lambda_{SS}) + \lambda_{SS}, & \text{either } i \text{ or } j \text{ is labeled,} \\ \eta(\lambda_{LL} - \lambda_{SS}) + \lambda_{SS}, & \text{both } i \text{ and } j \text{ are labeled.} \end{cases} \qquad (4)$$

The exact magnitudes of $\lambda_{SS}, \lambda_{SL}, \lambda_{LL}$ are given in Section 3.2. This model corresponds to a bidisperse system where $\eta$ controls the strength of the disorder. For theoretical simplicity, instead of using length parameter $a_{ij}$, we define an onsite parameter $t_i$. The variable $t_i$ takes a value of either 1 or 0, depending on whether the particle at position $\vec{r}_i$ is labeled or not. Using $t_i$ we can express $a_{ij}$ as follows

$$a_{ij} = \lambda_{SS} + \eta \left[ (t_i + t_j)(\lambda_{SL} - \lambda_{SS}) + t_i t_j (\lambda_{LL} + \lambda_{SS} - 2\lambda_{SL}) \right]. \qquad (5)$$

## 3 Numerical simulations

To test our theoretical predictions, we conduct simulations of an athermal, over-compressed triangular lattice (hcp) in two dimensions (2D) and a face-centered cubic (fcc) lattice in three dimensions (3D) with soft, frictionless particles with varying levels of particle size disorder. We employ periodic boundary conditions to account for boundary effects. Our focus is on states in which every particle achieves force balance, i.e., configurations corresponding to energy minima. To achieve this, we utilize the Fast Inertial Relaxation Engine (FIRE) algorithm as described in Ref. [65] to minimize the energy of the system.

Here the inter-particle separation is kept fixed at $R_0$ in the initial crystalline state. We initially consider a rectangular (2D) grid spacings of $R_0/2$ and $\sqrt{3}R_0/2$ along $x$ and $y$ directions respectively and cubic (3D) lattice with a grid spacing of $R_0$. To create a triangular/fcc arrangement, particles are placed on alternate grid points that satisfy the respective conditions, i.e., $n_x + n_y = 2n$ for a triangular lattice and $n_x + n_y + n_z = 2n$ for an fcc lattice, where $n$ is an integer. This technique is an extension of the one used in reference [66] for generating a hexagonal close packing in two dimensions. A square/cubic lattice has $4L^2/8L^3$ grid points in total (since there are $2L$ grid points along each coordinate axis), of which only half are occupied by particles, yielding the total number of particles, $N = 2L^2$ in 2D and $4L^3$ in 3D.

## 3.1 Harmonic model

We first consider the Harmonic model, where particles experience exclusively repulsive interactions and each grain interacts solely with its closest neighbors. In a two dimensional triangular lattice, this corresponds to six neighboring grains, while in a three dimensional fcc lattice, it corresponds to twelve neighboring grains. The degree of compression in the lattice is indicated by the packing fraction, which is set to $\phi = 0.92/0.96/0.98$ in 2D and $\phi = 0.80$ in 3D. This is in comparison to the marginally jammed triangular/fcc lattice with a packing fraction of $\phi_0 \approx 0.9069$ in 2D and $\phi_0 \approx 0.74$ in 3D. The interparticle spacing ($R_0$) is defined by the initial particle radius, which is set to $a_0 = 0.5$, and the packing fraction, calculated using the formula $R_0 = 2a_0(\phi_0/\phi)^{1/d}$, in the absence of any disorder. Here we have chosen bond stiffness/interaction strength as $K = 0.5$. The numerical results presented in this study are averaged over 200 different realizations of disordered states. These simulations were performed for system sizes of $N = 6400$ and $10000$ particles in two dimensions, and $N = 32000$ and $250000$ in three dimensions, with different strengths of particle size disorder ($\eta$).

## 3.2 LJ model

For the case of Lennard-Jones (LJ) interactions, every particle interacts with its neighboring particles located within a cutoff radius defined as $\left|\vec{r}_{ij}\right|/a_{ij} \leq 2.5$. In the absence of perturbation ($\eta = 0$), this condition implies that there are a total of 18 neighboring particles within the interaction range corresponding to each grain in the triangular lattice. Here we have chosen $K = 0.5$, with $\lambda_{SS} = 1, \lambda_{LL} = 1.4\lambda_{SS}$ and $\lambda_{SL} = 1.2\lambda_{SS}$ for the interaction potential. Similar to the Harmonic model, we have performed simulations for systems of size $N = 6400$ in $2D$. Since the volume associated with each particle is not defined, we define a number density ($\rho_N$) as our initial parameter instead of a packing fraction. Each number density, $\rho_N$ corresponds to different values of initial pressure ($P$) in the system. Here the results are presented for $P = 0/0.27/4.24$. To achieve the initial pressure we use a Berendsen barostat [67] which is implemented in the FIRE algorithm during the energy minimisation.

## 4 Elastic properties of near-crystalline packings

In this section, we derive the results that can be used to compute the relevant macroscopic elastic properties in a near-crystalline granular packing composed of soft particles. The fundamental property we are interested in is the global pressure, denoted by the symbol $P$ which can be written as

$$P = d^{-1}\sum_{\mu}\Sigma_{\mu\mu} = (dV)^{-1}\sum_{\mu,\langle ij\rangle} r_{ij}^{\mu}f_{ij}^{\mu}, \tag{6}$$

where $\Sigma_{\mu\mu}$ are the diagonal components of the global stress tensor where $d$ and $V$ represent the spatial dimension and total volume of the system respectively. Here $r_{ij}^{\mu}$ and $f_{ij}^{\mu}$ are the $\mu - th$ component of the relative displacement and force between the particles $i$ and $j$. The configurational averaged total pressure can be defined as

$$\langle P\rangle = d^{-1}\sum_{\mu}\langle\Sigma_{\mu\mu}\rangle = d^{-1}\sum_{\mu}\langle\Sigma_{\mu\mu}^{(0)}\rangle + \langle\delta\Sigma_{\mu\mu}^{(0)}\rangle, \tag{7}$$

where $\Sigma_{\alpha\beta}^0 = V^{-1}\sum_{\langle ij\rangle} r_{ij}^{\alpha(0)}f_{ij}^{\beta(0)}$ are the components of the global stress tensor of the crystalline system without the disorder. For a small magnitude of the disorder strength $\eta$, the average change in the global pressure is zero (i.e., $\langle\delta\Sigma_{\mu\mu}^{(0)}\rangle = 0$), which we will show in the

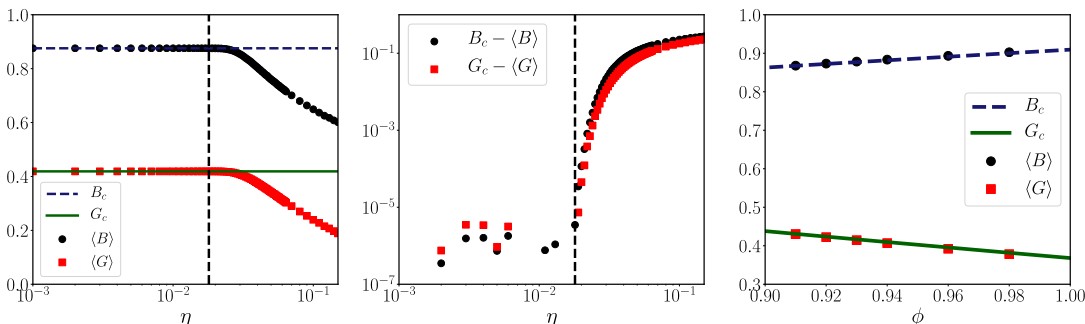

Figure 2: (*a*) Variation of configurational averaged bulk ($\langle B \rangle$) and shear modulus ($\langle G \rangle$) with polydispersity for a near-crystalline packing of soft particles in a system of size $N = 256$ with packing fraction, $\phi = 0.9269$ in two dimensions. (*b*) Difference between the elastic modulus of a crystal and that of a disordered crystal. The elastic properties of disordered crystals remain unaffected by the disorder up to a critical disorder strength ($\eta_c \approx 0.018$ for this packing fraction). (*c*) Variation of $\langle B \rangle$ and $\langle G \rangle$ with initial packing fraction for a fixed polydispersity, $\eta = 0.005$.

following section. For any regular arrangement of particles in $d$-dimension, we can write the relative distance between the neighboring particles and their corresponding forces as

$$\vec{r}_{ij}^{(0)} = R_0 \hat{r}_{ij}^0, \qquad \vec{f}_{ij}^{(0)} = \frac{K}{a_0} \left( 1 - \frac{R_0}{2a_0} \right) \hat{r}_{ij}^0, \tag{8}$$

where $R_0$ is the relative distance between two neighboring particles in the crystalline arrangement. Inserting these values, one can obtain the final form of the averaged net pressure as

$$\langle P \rangle = \frac{z_0 N K R_0}{2dV a_0} \left( 1 - \frac{R_0}{2a_0} \right) = \frac{z_0 K \rho_N^0}{d} \left( \frac{\phi_0}{\phi} \right)^{-1+1/d} \left( 1 - \left( \frac{\phi_0}{\phi} \right)^{1/d} \right). \tag{9}$$

Here $z_0$ is the coordination number of each grain and $\rho_N^0$ is the number density in the marginally jammed crystal. We can also represent the number density of an over-compressed crystal as, $\rho_N = N/V = \rho_N^0 (\phi/\phi_0)^{1/d}$. In the two dimensional triangular lattice, $z_0 = 6$ and $\rho_N^0 = 1/2\sqrt{3}a_0^2$. The system is subjected to an isotropic strain, where the box dimensions along all the Cartesian directions increase by a factor of $(1 + \epsilon)$. Consequently, the packing fraction $\phi$ changes to $\phi/(1 + \epsilon)^d$. The pressure $P$ of this isotropically strained system can be expressed as

$$\langle P' \rangle = \frac{z_0 N k R_0}{2dV a_0} \left( 1 - \frac{R_0}{2a_0} \right) = \frac{z_0 k \rho_N^0}{d(1+\epsilon)^{d-1}} \left( \frac{\phi_0}{\phi} \right)^{-1+1/d} \left( 1 - (1+\epsilon) \left( \frac{\phi_0}{\phi} \right)^{1/d} \right). \tag{10}$$

Here, $\epsilon \sim \delta V/2V$ is proportional to the volumetric strain applied to the system. The average bulk modulus ($\langle B \rangle$) for these near-crystalline systems can be obtained by finding the ratio of change in bulk pressure to the volumetric strain,

$$\langle B \rangle = \left| \frac{\delta \langle P \rangle}{\delta V/V} \right| = \lim_{\epsilon \to 0} \left| \frac{\langle P' \rangle - \langle P \rangle}{2\epsilon} \right| = \frac{z_0 k \rho_N^0}{2d} \left( \frac{\phi_0}{\phi} \right)^{-1+1/d} \left( (d-2) \left( \frac{\phi_0}{\phi} \right)^{1/d} - (d-1) \right). \tag{11}$$

In two dimensions, the average bulk modulus for a small magnitude of disorder in a crystalline packing can be written as $\langle B \rangle = \frac{\sqrt{3}K}{4a_0^2} (\phi/\phi_0)^{1/2}$, which we have also verified numerically for $\phi = \phi_0 + 0.02 = 0.9269$ as shown in Fig. 2. This demonstrates that the average elastic moduli of near-crystalline packings remain independent of the strength of the disorder below

a certain threshold, as indicated in previous studies [31]. This threshold coincides with the transition from a crystalline to a disordered crystal phase.

We next examine a near crystalline system subjected to shear along the $\alpha\beta$-plane with a shear amplitude denoted by $\gamma$. In response to this shear, particles undergo displacements, which comprises both affine and non-affine contributions. These displacements can then be used to compute the elastic moduli of the system. The major contribution from the non-affine displacement arises from the lack of inversion symmetry [68,69]. We measure the average degree of centrosymmetry ($F_{IS}$) (see Appendix D) in the configurations which give, $F_{IS} \sim 1$ i.e. $(1 - F_{IS} \sim \mathcal{O}(10^{-4}))$ for $\eta \leq 10^{-2}$. Therefore for such configurations, the contribution of the non-affinity to the elasticity is negligible [70]. Hence we only consider the affine part of the displacements for small shear amplitude, leading to the following expressions for the relative displacement and force components

$$r_{ij}^\alpha = R_0(\cos\theta_{ij}^0 + \gamma\sin\theta_{ij}^0), \qquad r_{ij}^\beta = R_0\sin\theta_{ij}^0, \qquad R_{ij} = \left(\sum_{\alpha=1}^{d}(r_{ij}^\alpha)^2\right)^{1/2}, \tag{12}$$

where $\theta_{ij}$ is the angle made by the projection of $r_{ij}$ on the $\alpha\beta$-plane to the $\alpha$-axis. Using the above relations, we can write the $\alpha\beta$-component of the stress tensor as

$$\langle\Sigma_{\alpha\beta}\rangle = \frac{1}{2V}\sum_i\left(\sum_j r_{ij}^\alpha f_{ij}^\beta\right) = k\rho_N^0\left(\frac{\phi_0}{\phi}\right)^{-1+1/d}\left(\sum_{j=1}^{z}\left(1 - \frac{R_{ij}}{2a_0}\right)(\cos\theta_{ij}^0\sin\theta_{ij}^0 + \gamma\sin^2\theta_{ij}^0)\right). \tag{13}$$

For a two dimensional triangular lattice (with $\alpha\beta \equiv xy$), the above equation simplifies to

$$\langle\Sigma_{xy}\rangle = \frac{3NkR_0}{2Va_0}\left[-\frac{R_0}{2a_0}\gamma + \frac{1}{\sqrt{3}}\left(\frac{\gamma - 1/\sqrt{3}}{\sqrt{1 + (\gamma - 1/\sqrt{3})^2}} + \frac{\gamma + 1/\sqrt{3}}{\sqrt{1 + (\gamma + 1/\sqrt{3})^2}}\right)\right]. \tag{14}$$

Considering a small shear amplitude, $\gamma$, we can do a linear approximation of the shear stress and take the ratio of change in the shear-stress to shear-strain to obtain the shear-modulus

$$G = \lim_{\gamma\to 0}\frac{\langle\Sigma_{xy}\rangle}{\gamma} \sim \frac{3NkR_0}{2Va_0}\left(\frac{R_0}{2a_0} - \frac{3}{4}\right) = \frac{\sqrt{3}k}{2a_0^2}\left(1 - \frac{3}{4}\left(\frac{\phi}{\phi_0}\right)^{1/2}\right). \tag{15}$$

Both the expressions for bulk and shear modulus for various packing fractions and particle size disorder are validated through numerical simulations in near-crystalline packings of soft particles as shown in Fig. 2. Any local fluctuations of pressure and shear stresses giving rise to local fluctuations in bulk and shear modulus are discussed in the later section. Given the planar bulk and shear moduli, one can obtain the expressions for planar Young's modulus and Poisson's ratio as

$$E = \frac{4BG}{B+G} = \frac{\sqrt{3}k}{a_0^2}\left(\frac{\phi}{\phi_0}\right)^{1/2}\left(\frac{4 - 3(\phi/\phi_0)^{1/2}}{4 - (\phi/\phi_0)^{1/2}}\right),$$
$$\nu = \frac{B-G}{B+G} = \frac{5(\phi/\phi_0)^{1/2} - 4}{4 - (\phi/\phi_0)^{1/2}}. \tag{16}$$

Similar techniques can be used in a system of particles interacting via long-ranged Lennard-Jones interactions in a near-crystalline packing with average particle separation between the nearest neighbor being $R_0$. The bulk and shear modulus for such a system can be computed as

$$B = \frac{24\sqrt{3}k}{R_0^8}\left(a - \frac{4b}{R_0^6}\right),$$
$$G = \frac{4\sqrt{3}k}{R_0^8}\left(a - \frac{5b}{R_0^6}\right), \tag{17}$$

where

$$a = \sum_{i=1}^{N} \frac{n_i}{m_i^6}, \text{ and } b = \sum_{i=1}^{N} \frac{n_i}{m_i^{12}}. \tag{18}$$

Here $n_i$ is the number of particles on the $i^{th}$ spherical cell and $m_i$ is the ratio of the distance of the $i^{th}$ cell from the central particle to $R_0$, i.e., $m_i = R_i/R_0$. In the $N \to \infty$ limit, we arrive at $a = 6.37588$ and $b = 6.00981$.

## 5 Displacement fields induced by microscopic disorder

In the previous section, we presented theoretical results related to the average stress tensor components in a nearly crystalline arrangement of soft particles, where we made the assumption that the average changes in local stress are negligible. The local stress components are proportional to the square of the interparticle distances of all neighboring particles within the cut-off distance. Consequently, in order to formulate the expressions for local stress, it is necessary to derive the displacements of individual particles in a disordered configuration. Below, we derive the displacement and force fields resulting from the introduction of disorder into a crystalline network.

In both Harmonic and LJ model, we begin with a crystalline packing of monodisperse particles in a fixed volume. In the short-ranged repulsive model we start with a finite over-compression whereas in the attractive LJ interaction model, the initial volume is fixed such that the initial pressure is set to $P = 0$. We then introduce disorder in the effective particle sizes i.e., $a_{ij}$ as given in Eqs. (2), (5). As a response to this disorder, the particles are displaced from their crystalline positions to maintain force balance, as

$$r_i^\mu = r_i^{\mu(0)} + \delta r_i^\mu. \tag{19}$$

Given that particle $j$ is one of the neighboring particles of particle $i$ in the initial crystalline lattice, the relative displacement between their positions can be expressed using the basis lattice vectors of the crystalline lattice as $\vec{r}_{ij}^{(0)} = \vec{r}_j^{(0)} - \vec{r}_i^{(0)} = \vec{\Delta}_j$. The discrete Fourier transform of the change in the relative displacement $\delta r_{ij}^\mu$ can be expressed as:

$$\mathcal{F}\left[\delta r_{ij}^\mu\right] = \sum_i e^{i\vec{r}_i^{(0)}.\vec{k}}\delta r_{ij}^\mu = \sum_i e^{i\vec{r}_i^{(0)}.\vec{k}}\left(\delta r_j^\mu - \delta r_i^\mu\right) = \left[e^{-i\vec{\Delta}_j.\vec{k}} - 1\right]\delta \tilde{r}^\mu(\vec{k}), \tag{20}$$

where $\delta \tilde{r}^\mu(\vec{k}) = \mathcal{F}[\delta r^\mu(\vec{r})]$, corresponds to the discrete Fourier transform of particle displacements from their crystalline positions. As a response to the disorder as well as the displacements in the particle positions, the forces $\vec{f}_{ij}$ acting between adjacent particles $i$ and $j$ also change. This variation can be expressed as a perturbation relative to the forces between these particles in the initial crystalline state, represented as

$$f_{ij}^\mu = f_{ij}^{\mu(0)} + \delta f_{ij}^\mu. \tag{21}$$

Every individual component of the excess force $\delta f_{ij}^\mu$ acting between particles $i$ and $j$ can be Taylor expanded up to first order about its value in the crystalline ground state, in terms of $\delta r_{ij}^\mu (= \delta r_j^\mu - \delta r_i^\mu)$ and $\delta a_{ij} (= \delta a_i + \delta a_j)$ as

$$\delta f_{ij}^\mu = \sum_\nu C_{ij}^{\mu\nu}\delta r_{ij}^\nu + C_{ij}^{\mu a}\delta a_{ij}, \tag{22}$$

where the first-order Taylor coefficients $C_{ij}^{\mu\nu}, C_{ij}^{\mu a}$ depend only on the form of the potential between the interacting particles and the initial crystalline arrangement. These coefficients can

be represented as, $C_{ij}^{\mu\nu} = \left(\partial f_{ij}^{\mu}/\partial r_{ij}^{\nu}\right)\Big|_{\{\vec{r}_{ij}^{(0)}, a_{ij}^{(0)}\}}$ and $C_{ij}^{\mu a} = \left(\partial f_{ij}^{\mu}/\partial a_{ij}\right)\Big|_{\{\vec{r}_{ij}^{(0)}, a_{ij}^{(0)}\}}$. For energy-minimized configurations, the force balance condition dictates that the net force acting on each particle $i$ is zero. This means that for all interacting neighbors $j$, the sum of the force deviations, denoted as $\delta\vec{f}_{ij}$, is equal to zero, expressed as $\sum_j \delta\vec{f}_{ij} = 0$. By applying this condition in the linear order expression for forces as given in Eq. (22), we obtain $d$-equations, where $d$ represents the dimension of the system, for each particle $i$. These equations are given as follows:

$$\sum_j \sum_\nu C_{ij}^{\mu\nu} \delta r_{ij}^\nu = -\sum_j C_{ij}^{\mu a} \delta a_{ij}\,,$$

$$\mathcal{P}_1 |\delta r\rangle = \mathcal{P}_2 |\delta a\rangle\,. \tag{23}$$

$\mathcal{P}_1$ is a matrix of the Taylor coefficients $C_{ij}^{\mu\nu}$ and $\mathcal{P}_2$ is a matrix containing elements $C_{ij}^{\mu a}$. Here, we have $Nd$ such equations corresponding to $Nd$-variables (displacement components). Since the system is translationally invariant, performing a discrete Fourier transform on the equation can convert the $Nd$-equations of $Nd$-variables into $d$-equations of $d$ variables. This simplification reduces the complexity of the problem and diagonalizes the large matrices $\mathcal{P}_1$ and $\mathcal{P}_2$. So the Fourier transform of the Eq. (23) leads to

$$\sum_\nu A^{\mu\nu}(\vec{k})\delta\tilde{r}^\nu(\vec{k}) = B^\mu(\vec{k})\,, \tag{24}$$

where

$$A^{\mu\nu}(\vec{k}) = \sum_j \left(1 - e^{-i\vec{k}.\vec{\Delta}_j}\right) C_{ij}^{\mu\nu}\,, \qquad B^\mu(\vec{k}) = -\mathcal{F}\left[\sum_j C_{ij}^{\mu a}\delta a_{ij}\right]. \tag{25}$$

In $d$-dimensions, $A^{\mu\nu}$ would be a $d \times d$ symmetric matrix. Here the expression for $A^{\mu\nu}$ and $B^\mu$ have the same form for both short and long-range models where the only difference lies in the number of interacting neighbor particles. Now we can obtain the displacement fields in Fourier space by inverting Eq. (24),

$$\delta\tilde{r}^\mu(\vec{k}) = \sum_\nu (A^{-1})^{\mu\nu}(\vec{k})B^\nu(\vec{k})\,. \tag{26}$$

Since $\delta\tilde{r}$ is expressed as the product of $A^{-1}$ and $B$, its inverse Fourier transform can be written as a convolution resulting in the displacement fields in real space, as shown below:

$$\delta r^\mu(\vec{r}) = \mathcal{F}^{-1}\left[\delta\tilde{r}^\mu(\vec{k})\right] = \frac{1}{N}\sum_{\vec{k}} \exp\left(-i\vec{k}.\vec{r}\right)\delta\tilde{r}^\mu(\vec{k})\,. \tag{27}$$

Here Eqs. (27) and (26) correspond to the displacement fields and their Fourier transform in the presence of particle size disorder. The exact expressions of these displacements are model dependent, and we discuss the two different scenarios in detail below.

## 5.1 Short-ranged repulsive harmonic interaction

The displacement fields for the Harmonic model has been studied extensively [32, 44, 45, 63] where the disorder is introduced in the particle radius, $\delta a_{ij} = \eta a_0(\zeta_i + \zeta_j)$. Next, putting this in Eq. (25) we get the expression for $B^\mu$,

$$B^\mu(\vec{k}) = -D^\mu(\vec{k})\delta\tilde{a}(\vec{k}) = -\eta a_0 D^\mu(\vec{k})\tilde{\zeta}(\vec{k})\,,$$

$$\text{where} \qquad D^\mu(\vec{k}) = \sum_j \left(1 + e^{-i\vec{k}.\vec{\Delta}_j}\right) C_{ij}^{\mu a}\,. \tag{28}$$

Here $\delta\tilde{a}(\vec{k}) = \mathcal{F}[\delta a(\vec{r})] = \eta a_0 \zeta(\vec{k})$, is the Fourier transform of $\delta a_i$. We have defined $|\vec{\Delta}_j| = R_0$ as the magnitude of the relative distance between grains in an over-compressed crystalline system. Now putting the expression of $B^\nu(\vec{k})$ in the expression of $\delta\tilde{r}^\mu(\vec{k})$ in Eq. (26), we obtain

$$\delta\tilde{r}^\mu(\vec{k}) = \underbrace{\left[-\sum_\nu (A^{-1})^{\mu\nu}(\vec{k}) D^\nu(\vec{k})\right]}_{\tilde{G}^\mu(\vec{k})} \delta\tilde{a}(\vec{k}) = \tilde{G}^\mu(\vec{k})\delta\tilde{a}(\vec{k}). \tag{29}$$

We can get the displacement fields by taking an inverse discrete Fourier transform as given in Eq. (27) as

$$\delta r^\mu(\vec{r}) = \sum_{\vec{r}'} G^\mu(\vec{r} - \vec{r}')\delta a(\vec{r}'),$$
$$\text{where} \quad G^\mu(\vec{r}) = \mathcal{F}^{-1}\left[\tilde{G}^\mu(\vec{k})\right]. \tag{30}$$

## 5.2 Attractive Lennard-Jones interaction with cut-off

The above-mentioned formulation for displacement fields can also be extended to any sort of interaction where every grain can interact with 6 or more neighbors depending on the interaction cut-off. For example, this cutoff is $|\vec{r}_{ij}|/a_{ij} \leq 2.5$ for the LJ model, where the microscopic disorders are incorporated into the bond distances as

$$\delta a_{ij} = \eta\left[(t_i + t_j)(\lambda_{SL} - \lambda_{SS}) + t_i t_j(\lambda_{LL} + \lambda_{SS} - 2\lambda_{SL})\right]. \tag{31}$$

Therefore, $B^\mu(\vec{k})$ has the form

$$B^\mu(\vec{k}) = \eta\left[(\lambda_{SS} - \lambda_{SL})\tilde{D}^\mu(\vec{k})\tilde{t}(\vec{k}) - \frac{(\lambda_{LL} + \lambda_{SS} - 2\lambda_{SL})}{N}\sum_{\vec{k}'}\left[\tilde{t}(\vec{k}')\tilde{t}(\vec{k} - \vec{k}')\tilde{D}^\mu(\vec{k} - \vec{k}')\right]\right]. \tag{32}$$

In the above expression, $\tilde{D}^\mu(\vec{k})$ has the same expression as given in Eq. (28) with $j$ going from $1 - 18$ for all the interacting neighbors within the range $|\vec{r}_{ij}|/a_{ij} \leq 2.5$. The only difference in the expression of $B^\mu$ in LJ model to that of the Harmonic model is that the magnitude of $\vec{\Delta}_j$s are not constant for all the interacting neighbors. In this study, we have chosen, $\lambda_{SL} = (\lambda_{SS} + \lambda_{LL})/2$ for numerical simulations, which simplifies the problem by removing the nonlinear term in the above expression for $B^\mu(\vec{k})$. In this approximation we can write, $\delta a_i = \eta(\lambda_{SL} - \lambda_{SS})t_i$. Now the expressions for $B^\mu(\vec{k})$ and $\delta\tilde{r}^\mu(\vec{k})$ in Eq. (26) can be written as

$$B^\mu(\vec{k}) = -D^\mu(\vec{k})\delta\tilde{a}(\vec{k}) = -\eta(\lambda_{SL} - \lambda_{SS})D^\mu(\vec{k})\tilde{t}(\vec{k}),$$
$$\delta\tilde{r}^\mu(\vec{k}) = \underbrace{\left(-\sum_\nu (A^{-1})^{\mu\nu}(\vec{k}) D^\nu(\vec{k})\right)}_{\tilde{G}^\mu(\vec{k})} \delta\tilde{a}(\vec{k}). \tag{33}$$

where $\delta\tilde{a}(\vec{k}) = \eta(\lambda_{SL} - \lambda_{SS})\tilde{t}(\vec{k})$.

# 6 Stress correlations induced by microscopic disorder

In this section, we focus on the fluctuations and correlations of the local stress tensor components. Using the linear order displacement fields derived in the earlier section, we can compute the stress correlations using a similar perturbation expansion for the minimally polydisperse

system. For any athermal jammed system, the components of the global stress tensor can be represented as

$$\Sigma_{\alpha\beta} = V^{-1} \sum_{\langle ij \rangle} r_{ij}^{\alpha} f_{ij}^{\beta}, \tag{34}$$

where $r_{ij}^{\alpha} = r_{ij}^{\alpha(0)} + \delta r_{ij}^{\alpha}$ and $f_{ij}^{\beta} = f_{ij}^{\beta(0)} + \delta f_{ij}^{\beta}$. Here $r_{ij}^{\mu(0)}$ refer to the $\mu$-th component of the interparticle distance whereas $f_{ij}^{\mu(0)}$ denote the $\mu$-th component of the force acting between particle $i$ and $j$ in the crystalline lattice without any microscopic disorder. In a system with small particle size polydispersity ($\eta$), both the deviation of the particle positions ($|\delta \vec{r}|$) from the crystalline positions and the change in inter-particle forces ($|\delta \vec{f}|$) are in the order of $\delta a \sim \eta$. For small values of $\eta$, we can neglect higher-order terms in the expression for global stress which leads to

$$\Sigma_{\alpha\beta} \sim V^{-1} \sum_{\langle ij \rangle} \left( r_{ij}^{\alpha(0)} f_{ij}^{\beta(0)} + r_{ij}^{\alpha(0)} \delta f_{ij}^{\beta} + \delta r_{ij}^{\alpha} f_{ij}^{\beta(0)} \right). \tag{35}$$

Therefore the incremental change in the global stress (i.e $\delta\Sigma = \Sigma - \Sigma^{(0)}$) due to the introduction of disorder in the crystalline system can be written as

$$\delta\Sigma_{\alpha\beta} = V^{-1} \sum_i \underbrace{\left[ \sum_j (r_{ij}^{\alpha(0)} \delta f_{ij}^{\beta} + \delta r_{ij}^{\alpha} f_{ij}^{\beta(0)}) \right]}_{\delta\sigma_{\alpha\beta}(\vec{r}_i)}. \tag{36}$$

The above expression represents the net change in global stress as a linear combination of $\delta\sigma(\vec{r}_i^0)$, which we define as the change in local stress at the lattice position $\vec{r}_i^{(0)}$, and can be expressed as follows:

$$\delta\sigma_{\alpha\beta}(\vec{r}_i^0) = \sum_j (r_{ij}^{\alpha(0)} \delta f_{ij}^{\beta} + \delta r_{ij}^{\alpha} f_{ij}^{\beta(0)}) = \sum_j \left[ \Delta_j^{\alpha} \sum_{\nu} C_{ij}^{\beta\nu} \delta r_{ij}^{\nu} + \Delta_j^{\alpha} C_{ij}^{\beta a} \delta a_{ij} + \delta r_{ij}^{\alpha} f_{ij}^{\beta(0)} \right]. \tag{37}$$

In the above expression, $\delta\sigma_{\alpha\beta}(\vec{r}_i)$ is a linear summation of particle displacements and change in radii with coefficients. As we have demonstrated earlier, the Fourier transform of $\delta r_{ij}^{\alpha}$ and $\delta a_{ij}$ have simple relationships due to the translational invariance of the system, as shown in Eqs. (29) and (33). Therefore, we can simplify the problem by performing a discrete Fourier transform of Eq. (37) which leads to

$$
\begin{aligned}
\delta\tilde{\sigma}_{\alpha\beta}(\vec{k}) &= \mathcal{F}\left[ \delta\sigma_{\alpha\beta}(\vec{r}_i^0) \right] \\
&= \sum_i e^{i\vec{r}_i^0 \cdot \vec{k}} \delta\sigma_{\alpha\beta}(\vec{r}_i^0) \\
&= \sum_j \left[ \Delta_j^{\alpha} \sum_{\nu} C_j^{\beta\nu} \left[ -1 + F_j(\vec{k}) \right] \delta\tilde{r}^{\nu}(\vec{k}) + \Delta_j^{\alpha} C_j^{\beta a} \left[ 1 + F_j(\vec{k}) \right] \delta\tilde{a}(\vec{k}) \right. \\
&\quad \left. + \left[ -1 + F_j(\vec{k}) \right] \delta\tilde{r}^{\alpha}(\vec{k}) f_j^{\beta(0)} \right],
\end{aligned}
\tag{38}
$$

where $F_j(\vec{k}) = \exp\{-i\vec{\Delta}_j \cdot \vec{k}\}$. We can further simplify the above expression by replacing $\delta\tilde{r}^{\nu}(\vec{k})$ by $\tilde{G}^{\nu}(\vec{k})\delta\tilde{a}(\vec{k})$ as given in Eqs. (29) and (33), to arrive at

$$\delta\tilde{\sigma}_{\alpha\beta}(\vec{k}) = S_{\alpha\beta}(\vec{k})\delta\tilde{a}(\vec{k}), \qquad \text{where}$$

$$S_{\alpha\beta}(\vec{k}) = \sum_j \left[ \left[ 1 + F_j(\vec{k}) \right] C_j^{\beta a} \Delta_j^{\alpha(0)} + \left[ -1 + F_j(\vec{k}) \right] \left( \sum_{\nu} \Delta_j^{\alpha(0)} C_j^{\beta\nu} \tilde{G}^{\nu}(\vec{k}) + f_j^{\beta(0)} \tilde{G}^{\alpha}(\vec{k}) \right) \right]. \tag{39}$$

The sum over $j$ pertains to all neighboring particles, i.e., all the particles that interact with the central particle in the crystalline state without the disorder. Here, $S_{\alpha\beta}$ represents the Fourier transform of the Green's function for the change in local stress components. Next, we can obtain the change in local stresses in real space as a convolution by performing an inverse Fourier transform of Eq. (39). This yields

$$\delta\sigma_{\alpha\beta}(\vec{r}) = \mathcal{F}^{-1}\left[\delta\tilde{\sigma}_{\alpha\beta}(\vec{k})\right] = \sum_{\vec{r}'} S_{\alpha\beta}(\vec{r}-\vec{r}')\delta a(\vec{r}'), \tag{40}$$

where $S_{\alpha\beta}$ is Green's function for the change in local stresses. Next, we can write the form for the Fourier transform of the change in the local pressure as

$$\delta\tilde{P}(\vec{k}) = d^{-1}\sum_{\alpha=1}^{d}\delta\tilde{\sigma}_{\alpha\alpha}(\vec{k}) = d^{-1}\delta\tilde{a}(\vec{k})\sum_{\alpha}S_{\alpha\alpha}(\vec{k}). \tag{41}$$

Since the Fourier transform of the change in the local stresses is linearly proportional to $\delta\tilde{a}(\vec{k})$, we can also derive the configurational average of the local stress and pressure correlations as

$$\langle\delta\tilde{\sigma}_{\alpha\beta}(\vec{k})\delta\tilde{\sigma}_{\mu\nu}(\vec{k}')\rangle = \langle\delta\tilde{a}(\vec{k}).\delta\tilde{a}(\vec{k}')\rangle S_{\alpha\beta}(\vec{k})S_{\mu\nu}(\vec{k}'),$$

$$\langle\delta\tilde{P}(\vec{k})\delta\tilde{P}(\vec{k}')\rangle = \frac{\langle\delta\tilde{a}(\vec{k}).\delta\tilde{a}(\vec{k}')\rangle}{d^2}\sum_{\alpha,\beta}S_{\alpha\alpha}(\vec{k})S_{\beta\beta}(\vec{k}'). \tag{42}$$

Using the translational invariance of the system, the configurational average of the microscopic correlations of $\delta\tilde{a}(\vec{k})$ between two points $\vec{k}$ and $\vec{k}'$ in Fourier space can be written as

$$\langle\delta\tilde{a}(\vec{k}).\delta\tilde{a}(\vec{k}')\rangle = \frac{N\eta^2}{48}\delta_{\vec{k},-\vec{k}'} \qquad \text{(Harmonic)}$$

$$= \frac{N\eta^2(\lambda_{SL}-\lambda_{SS})^2}{4}\delta_{\vec{k},-\vec{k}'} \qquad \text{(LJ)}. \tag{43}$$

In the $|\vec{k}| \to 0$ limit, $\langle\delta\tilde{\sigma}_{\alpha\beta}(\vec{k})\delta\tilde{\sigma}_{\mu\nu}(-\vec{k})\rangle$ becomes independent of the magnitude of $|\vec{k}|$ and only has an angular dependence which we represent as

$$C_{\alpha\beta\mu\nu}(\theta) = \lim_{|\vec{k}|\to 0}\langle\delta\tilde{\sigma}_{\alpha\beta}(\vec{k}).\delta\tilde{\sigma}_{\mu\nu}(-\vec{k})\rangle. \tag{44}$$

The observed stress correlations are therefore anisotropic in the $k \to 0$ limit, corresponding to a pinch-point singularity at $k = 0$ [43]. Due to the finite system size, and to avoid effects introduced by the periodic boundaries, we have integrated the stress correlations in Fourier space in the narrow window of $0.5 \le |\vec{k}| \le 1.5$. The integrated stress correlations in a small window of $k \in [k_{min}, k_{max}]$ near $k \to 0$ can be expressed as

$$\bar{C}_{\alpha\beta\mu\nu}(\theta) = \int_{k_{min}}^{k_{max}} dk\langle\delta\tilde{\sigma}_{\alpha\beta}(k,\theta)\delta\tilde{\sigma}_{\mu\nu}(k,\pi+\theta)\rangle. \tag{45}$$

In real space, this translates to integrating the stress correlations at intermediate to large lengthscales. The angular dependence of the integrated correlations are plotted in Fig. 5 for both Harmonic and LJ model.

## 6.1 Local stress fluctuations

We can express the correlation of the excess local stress between two points $\vec{r}$ and $\vec{r}'$ in real space using the expression for stress correlation in $k$-space given in Eq. (42) as

$$\langle\delta\sigma_{\mu\nu}(\vec{r})\delta\sigma_{\mu\nu}(\vec{r}')\rangle = \sum_{\vec{k},\vec{k}'}\frac{\langle\delta\tilde{\sigma}_{\mu\nu}(\vec{k})\delta\tilde{\sigma}_{\mu\nu}(\vec{k}')\rangle}{N^2}e^{-i(\vec{k}.\vec{r}+\vec{k}'.\vec{r}')} = \frac{\eta^2}{48N}\sum_{\vec{k}}S_{\mu\nu}(\vec{k})S_{\mu\nu}(-\vec{k})e^{-i(\vec{r}-\vec{r}').\vec{k}}. \tag{46}$$

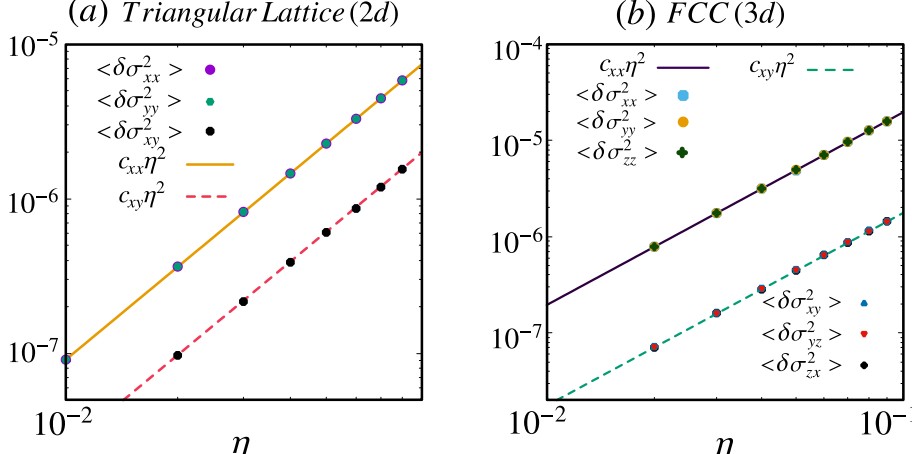

Figure 3: Local stress fluctuation with increasing polydispersity for ($a$) disordered triangular lattice (2d) with repulsive harmonic particles of system size, $N = 256$ and initial packing fraction, $\phi = 0.92$ and ($b$) disordered fcc lattice (3d) with $N = 4000$ and $\phi = 0.80$. Here all the numerical stress fluctuation varies as $c_{\alpha\beta}\eta^2$, where $c_{\alpha\beta} = \frac{1}{48N}\sum_{\vec{k}} S_{\alpha\beta}(\vec{k})S_{\alpha\beta}(-\vec{k})$.

Therefore the local stress fluctuations at the same site can be written as

$$\langle\delta\sigma^2_{\mu\nu}(\vec{r})\rangle = \frac{\eta^2}{48N}\sum_{\vec{k}} S_{\alpha\beta}(\vec{k})S_{\mu\nu}(-\vec{k})\,. \tag{47}$$

The theoretical prediction for local stress fluctuations as a function of increasing polydispersity is presented in Figure 3($a$) and ($b$) for a two dimensional triangular lattice and a three dimensional fcc lattice, respectively. These theoretical stress fluctuations align perfectly with the numerical results.

## 6.2 Stress correlations in two dimensional systems

In the case of the two dimensional Harmonic model, every grain has six neighbors in the near-crystalline system. The expressions for the Green's functions $S_{\alpha\beta}$, as defined in Eq. (39), depend solely on the nearest neighbor arrangement. Our analytic results demonstrate that these Green's functions in Fourier space have no radial dependence at small values of $|\vec{k}|$, corresponding to larger lengthscales in real space. Keeping only the first term in the Taylor expansion we can write these Green's functions as

$$
\begin{aligned}
S_{xx} &= \frac{\mathcal{C}(R_0,K)}{|\vec{k}|^2}\left[\left(\frac{R_0}{a_0}-1\right)k_y^2 + \left(2-\frac{R_0}{a_0}\right)k_x^2\right] + \mathcal{O}(k^2)\,,\\
S_{yy} &= \frac{\mathcal{C}(R_0,K)}{|\vec{k}|^2}\left[\left(\frac{R_0}{a_0}-1\right)k_x^2 + \left(2-\frac{R_0}{a_0}\right)k_y^2\right] + \mathcal{O}(k^2)\,,\\
S_{xy} &= \frac{\mathcal{C}(R_0,K)}{|\vec{k}|^2}\left[\left(3-\frac{2R_0}{a_0}\right)k_x k_y\right] + \mathcal{O}(k^2)\,,\\
\delta\tilde{P} &= \frac{\mathcal{C}(R_0,K)}{2}\delta\tilde{a}(\vec{k})\left[1 + f_1|\vec{k}|^2 + f_2|\vec{k}|^2 + \ldots\right]\,,
\end{aligned}
\tag{48}
$$

where

$$\mathcal{C}(R_0,K) = \frac{6KR_0(R_0-a_0)}{(2R_0-a_0)a_0^2}\,,\quad f_1 = -(a_0/3)^2\cos^2 3\theta\,,\quad\text{and}\quad f_2 = f_1^2\,. \tag{49}$$

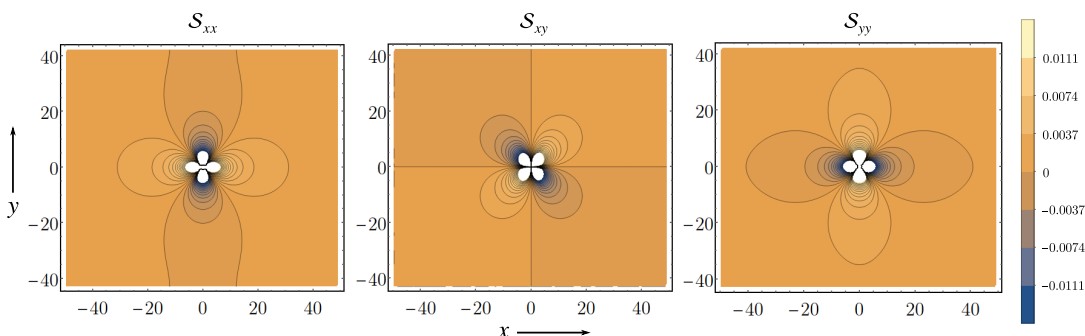

Figure 4: Change in local stress due to the introduction of disorder in a single particle (i.e., the Green's function for change in the local stress) in a 2D near-crystalline packing (HCP) of soft particles. Here figure (a), (b), and (c) correspond to $\delta\sigma_{xx}(\vec{r})$, $\delta\sigma_{xy}(\vec{r})$ and $\delta\sigma_{yy}(\vec{r})$ for $\delta a(\vec{r}=0)=1$ and $\delta a(\vec{r}\neq 0)=0$ for every other grain.

Here the functions $f_1$ and $f_2$ in the expression for $\delta\tilde{P}$ are influenced by the initial packing fraction and the orientation of the reciprocal lattice vector $\vec{k}$. The above expressions for Green's functions can be reformulated in a polar coordinate system to describe behavior at large lengthscales as follows.

$$
\lim_{|\vec{k}|\to 0} S_{xx}(|\vec{k}|,\theta) \sim \frac{\mathcal{C}(R_0,K)}{2}\left(1-\left(\frac{2R_0}{a_0}-3\right)\cos(2\theta)\right),
$$
$$
\lim_{|\vec{k}|\to 0} S_{yy}(|\vec{k}|,\theta) \sim \frac{\mathcal{C}(R_0,K)}{2}\left(1+\left(\frac{2R_0}{a_0}-3\right)\cos(2\theta)\right), \tag{50}
$$
$$
\lim_{|\vec{k}|\to 0} S_{xy}(|\vec{k}|,\theta) \sim -\frac{\mathcal{C}(R_0,K)}{2}\left(\frac{2R_0}{a_0}-3\right)\sin(2\theta).
$$

All the Green's function above and their correlations show anisotropic behavior with an angular periodicity of $\pi$. Performing an inverse Fourier transform on the above expression yields the behavior of these Green's functions at large lengthscales in real space, which is presented below as

$$
S_{xx}(\vec{r})=-S_{yy}(\vec{r}) \sim -\frac{\mathcal{C}(R_0,K)}{2}\left(\frac{2R_0}{a_0}-3\right)\frac{\cos(2\theta)}{|\vec{r}|^2},
$$
$$
S_{xy}(\vec{r})=S_{yx}(\vec{r}) \sim -\frac{\mathcal{C}(R_0,K)}{2}\left(\frac{2R_0}{a_0}-3\right)\frac{\sin(2\theta)}{|\vec{r}|^2}. \tag{51}
$$

Fig. 4 displays all the components of the Green's functions at large lengthscales whose functional forms are given in Eq. (51). The long-ranged two dimensional LJ model also exhibits similar behavior at large lengthscales. The Green's functions as well as stress correlations in LJ model have equal angular behavior as that of the Harmonic model as plotted in Figs. 4 and 5, only difference lying in the magnitude of the stress fluctuations which depends on the initial macroscopic properties of these systems like global pressure, box size. In the two dimensional Harmonic model, all six unique stress correlations as well as the pressure correlation in Fourier space at small magnitudes of $\vec{k}$ can be represented using the expressions given in Eqs. (42) and (48). For example, the correlation of change in local pressure at large lengthscales ($k\to 0$) can be expressed as

$$
\mathcal{P}(R_0,\eta)=\lim_{k\to 0}\left\langle \delta\tilde{P}(\vec{k}).\delta\tilde{P}(-\vec{k})\right\rangle \sim N\langle\delta a^2\rangle\frac{\mathcal{C}^2(R_0,K)}{4}, \tag{52}
$$

which is a constant for a given packing of particles. The higher-order terms in the Taylor expansion of pressure correlation in Fourier space reveal the anisotropic crystalline nature

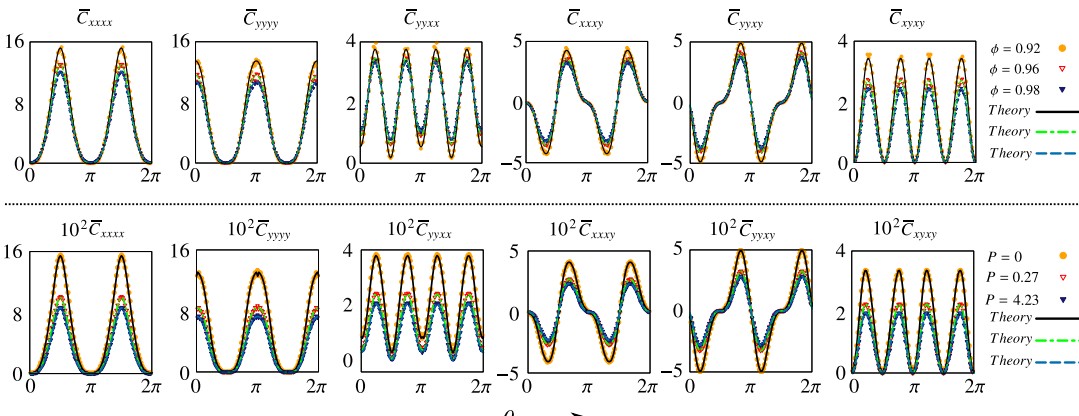

Figure 5: Angular dependence of stress correlations in Fourier space integrated in a small window of $|\vec{k}|$ i.e $\bar{C}_{\alpha\beta\mu\nu}(\theta) = \int_{k_{min}}^{k_{max}} dk \langle \delta\tilde{\sigma}_{\alpha\beta}(k,\theta)\delta\tilde{\sigma}_{\mu\nu}(k,\pi+\theta)\rangle$ for different initial over-compression (Harmonic)/pressure (LJ) with disorder strength of $\eta = 0.005$. The first row represents all the six distinct correlations for the Harmonic model and the second row corresponds to the LJ model. Here the solid and dashed lines correspond to the theoretical results where points correspond to the numerical data for the two different models. Here we have chosen $k_{min} = 0.1$ and $k_{max} = 1.0$ and system size $N = 6400$ for both the models.

of the inherent lattice. However, the significance of higher-order terms becomes apparent at smaller length scales, where lattice symmetry becomes a dominant factor.

In Fig. 5 we show an exact match between the stress correlations obtained from numerical simulations and the predictions from the microscopic theory for both short-ranged Harmonic and long-ranged LJ model in two dimensions. The aforementioned correlations are applicable to systems that possess a finite average pressure ($R_0 < 2a_0$). However, in the limit where the average pressure is zero ($\langle P \rangle \to 0$), i.e., as $R_0$ approaches $2a_0$, the results obtained from the VCTG framework [18,43] for amorphous materials lacking any crystalline symmetries are reproduced i.e.

$$\lim_{R_0 \to 2a_0} \left( \lim_{k \to 0} S_{\alpha\beta} \right) = \frac{\mathcal{C}(2a_0, K)}{(-1)^{1-\delta_{\alpha\beta}} k^2} \prod_{i=\alpha,\beta} \left( \sqrt{k^2 - k_i^2} \right), \tag{53}$$

which can be used to write the correlations between the different components of the stresses as

$$C_{\alpha\beta\mu\nu} = \frac{N\eta^2}{48} \left[ \lim_{R_0 \to 2a_0} \left( \lim_{k \to 0} S_{\alpha\beta}(\vec{k}) S_{\mu\nu}(-\vec{k}) \right) \right] = \frac{4\mathcal{P}(2a_0, \eta)}{(-1)^{2-\delta_{\alpha\beta}-\delta_{\mu\nu}} k^4} \prod_{i=\alpha,\beta,\mu,\nu} \left( \sqrt{k^2 - k_i^2} \right). \tag{54}$$

where $\mathcal{C}(2a_0, K) = 4K/a_0$ and $\mathcal{P}(2a_0, \eta) = \eta^2 K^2 / 12a_0^2$. In the same $R_0 \to 2a_0$ limit, the functions in Eq. (48) for change in local pressure have the following form i.e., $f_1 = -(a_0/3)^2 \cos^2 3\theta$, $f_2 = f_1^2$ and so on. For a finite system with particle radius $a_0$, the maximum magnitude of $k$ is $\pi/a_0$. So $|f_1 k^2| \leq 1$ for all values of $k$, except near the boundary of the $1^{st}$ Brillouin zone. So the change in the local pressure due to particle size defect for $k \ll \pi/a_0$ can be written as,

$$\lim_{R_0 \to 2a_0} \delta\tilde{P}(\vec{k}) = \frac{\mathcal{C}(2a_0, K)\delta\tilde{a}(\vec{k})}{2}(1 - f_1 k^2 + f_1^2 k^4 + ...)$$

$$\sim \frac{\mathcal{C}(2a_0, K)\delta\tilde{a}(\vec{k})}{2(1 + f_1 k^2)} = \frac{\mathcal{C}(2a_0, K)\delta\tilde{a}(\vec{k})}{2\left(1 + (a_0/3)^2 k^2 \cos^2 3\theta\right)}. \tag{55}$$

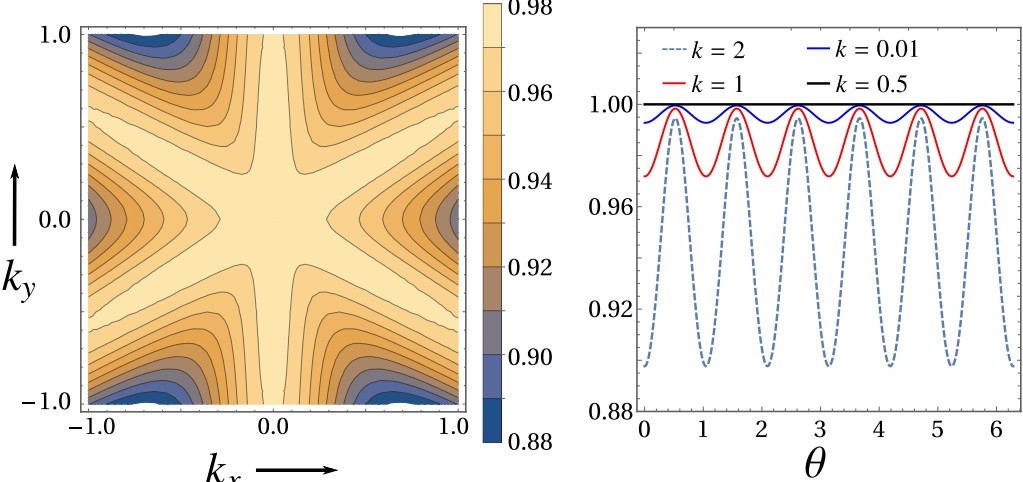

Figure 6: Correlation in the change in local pressure in Fourier space $(\langle \delta \tilde{P}(\vec{k}) \delta \tilde{P}(-\vec{k}) \rangle / \langle \delta a^2 \rangle)$ due to introduction of a single particle disorder in a crystalline packing of soft particles with $\phi = 0.92$.

Using the above approximation, we can calculate the correlation of local pressure which have the sixfold symmetry (i.e $\cos^2 3\theta$) for $|\vec{k}| > 0$, which we have shown in Fig. 6 for a single particle disorder. In the zero average pressure limit, the findings for the stress correlations exhibit unexpected universal behavior. This universal behavior is characterized by the observation of similar anisotropic stress correlations at large length scales across various jammed athermal packings [17, 18, 43, 58, 59].

## 6.3 Comparision with amorphous systems at large lengthscales

Recently the stress correlations in fully amorphous packings have been successfully predicted within a field-theoretic framework [18, 43]. This Vector Charge Theory of "emergent" elasticity is defined by the following equations:

$$\partial_i \Sigma_{ij} = f_j \,, \tag{56}$$

$$E_{ij} = \frac{1}{2}(\delta_i \psi_j + \delta_j \psi_i) \,, \tag{57}$$

$$\sigma_{ij} = (\delta_{ijkl} + \chi_{ijkl})E_{kl} = \Lambda^{-1}_{ijkl}E_{kl} \,. \tag{58}$$

The stress tensor field is represented by $\sigma$, and in this context, $E$ serves a role that is similar to that of the strain field in canonical elasticity theory. The emergent elasticity modulus tensor is defined as $\Lambda$, and the equations bear a notable resemblance to those of canonical linear elasticity. The components of the $\Lambda$ tensor can be interpreted as "emergent" elastic moduli. By utilizing the tensor gauge theory for polarizable isotropic media, all 6 distinct stress correlations at small magnitudes of $\vec{k}$ can be obtained which has the same form as the ones in Eq. (54) for near-crystalline systems in the $P \to 0$ limit with $\mathcal{P}(2a_0, \eta)$ replaced by a constant $K_{2D}$ that depends on the elastic properties of the system [18, 43]. So for the finite pressure, the stress correlations for the near-crystalline systems can be represented as a summation of the stress correlation for an isotropic amorphous system (from the VCTG framework) and the non-isotropic part of the stress correlation which contains the information about the crystalline symmetry i.e.

$$C_{\alpha\beta\mu\nu} = C^t_{\alpha\beta\mu\nu} + d_{\alpha\beta\mu\nu} \,, \tag{59}$$

where $C^t_{\alpha\beta\mu\nu}$, represents the stress correlations obtained from the VCTG framework. This finite shift $d_{\alpha\beta\mu\nu}$, can also be seen from the numerically obtained correlations as plotted in Fig. 5 which can not be explained in the VCTG framework. Therefore for an overcompressed disordered crystal with finite average pressure, the angular behavior of stress correlations show an additional anisotropic term which depend on the system pre-stress. For small pre-stress i.e., $R_0 \rightarrow 2a_0$, we have $|d_{\alpha\beta\mu\nu}/C_{\alpha\beta\mu\nu}| \rightarrow 0$, and consequently numerically the stress correlations are indistinguishable to that of the amorphous systems. The exact expressions for the angular dependence of the above correlation functions are detailed in Appendix B.

## 6.4 Continuum limit

In the small $|\vec{k}|$ limit, using the expression for $S_{\alpha\beta}(\vec{k})$ as given in Eq. (48) we can rewrite the change in the local stresses in the Fourier space as

$$\delta\tilde{\sigma}_{\alpha\beta} = \left[ k_y^2 \phi_{\alpha\beta 1} + k_x^2 \phi_{\alpha\beta 2} - k_x k_y \phi_{\alpha\beta 3} \right] |\vec{k}|^{-2} \delta\tilde{a}(\vec{k}). \tag{60}$$

Using Voigt notation [71], we can replace $xx = 1, yy = 2, xy = 3$. Since the stress tensor has three independent components in two dimensions, we can represent the Fourier transform of their change in the following matrix form,

$$\lim_{k \to 0} \begin{bmatrix} \delta\tilde{\sigma}_1(\vec{k}) \\ \delta\tilde{\sigma}_2(\vec{k}) \\ \delta\tilde{\sigma}_3(\vec{k}) \end{bmatrix} = \underbrace{\left( \frac{\delta\tilde{a}(\vec{k})}{|\vec{k}|^2} \right)}_{\tilde{\Psi}(\vec{k})} \underbrace{\begin{bmatrix} \phi_{11} & \phi_{12} & \phi_{13} \\ \phi_{21} & \phi_{22} & \phi_{23} \\ \phi_{31} & \phi_{32} & \phi_{33} \end{bmatrix}}_{\hat{\Phi}} \cdot \underbrace{\begin{bmatrix} k_y^2 \\ k_x^2 \\ -k_x k_y \end{bmatrix}}_{|A(\vec{k})\rangle}, \tag{61}$$

$$\left| \delta\tilde{\sigma}(\vec{k}) \right\rangle = \hat{\Phi} \left( \tilde{\Psi}(\vec{k}) \left| A(\vec{k}) \right\rangle \right),$$

where

$$\begin{aligned} \phi_{11} = \phi_{22} &= \mathcal{C}(R_0, K)(R_0/a_0 - 1), \\ \phi_{33} &= \mathcal{C}(R_0, K)(2R_0/a_0 - 3), \\ \phi_{12} = \phi_{21} &= \mathcal{C}(R_0, K)(2 - R_0/a_0), \\ \phi_{13} = \phi_{23} = \phi_{31} = \phi_{32} &= 0. \end{aligned} \tag{62}$$

Given that $\delta\tilde{\sigma}_i(\vec{k})$ characterizes the Fourier transform of the change in local stresses as $|\vec{k}| \rightarrow 0$, its inverse-Fourier transform reveals the changes in local stresses at large lengthscales due to the defect at the origin. Specifically, this pertains to the coarse-grained local stress fluctuations in real space, which can be expressed as:

$$\delta\sigma_i(\vec{r}) = \left[ \phi_{i1}\partial_y^2 + \phi_{i2}\partial_x^2 - \phi_{i3}\partial_x\partial_y \right] \Psi(\vec{r}), \tag{63}$$

where

$$\Psi(\vec{r}) = \frac{1}{(2\pi)^2} \int d^2k\, e^{i\vec{r}\cdot\vec{k}} \left( \frac{\delta\tilde{a}(\vec{k})}{|\vec{k}|^2} \right). \tag{64}$$

The summation in the above equation is performed over reciprocal lattice points in a triangular lattice configuration. The function $\Psi(\vec{r})$ is a non-isotropic field contingent upon the symmetry of the lattice. In an isotropic system, the force balance criterion is:

$$\partial_i \sigma_{ij}(\vec{r}) = f^j(\vec{r}) = 0. \tag{65}$$

In the case of an amorphous system, we can establish an isotropic field $\Psi'(\vec{r})$, which lacks any lattice symmetry, yet meets the force balance condition as expressed in Eq. (65), in a manner such that

$$\delta\sigma'_{ij}(\vec{r}) = \epsilon_{ia}\epsilon_{jb}\partial_a\partial_b\Psi'(\vec{r}). \tag{66}$$

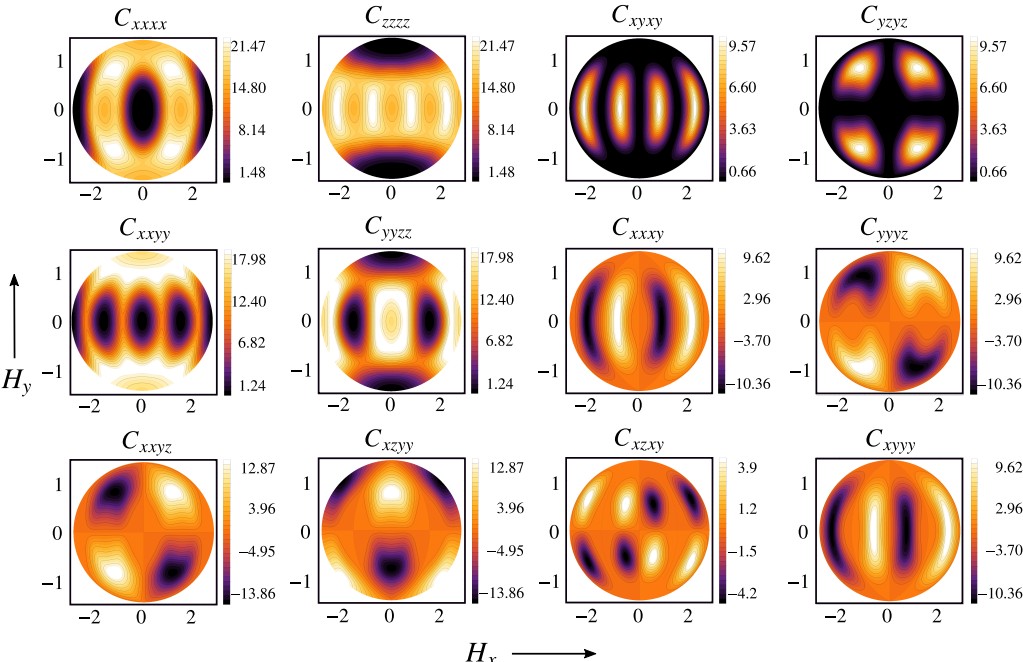

Figure 7: Angular variation of the change in local stress correlation in Fourier space due to particle size disorder in a 3D near-crystalline (fcc) system as $|\vec{k}| \to 0$. Here $C_{\alpha\beta\mu\nu}(\theta, \phi) = \lim_{|\vec{k}|\to 0} \left\langle \delta\tilde{\sigma}_{\alpha\beta}(\vec{k}).\delta\tilde{\sigma}_{\mu\nu}(-\vec{k}) \right\rangle = \frac{N\eta^2}{48} S_{\alpha\beta} S_{\mu\nu}$, where exact expressions for $S_{\alpha\beta}$ and its angular variations are given in Eqs. (A.4), (A.5) and (A.6). Here we have plotted correlations the in Hammer projection as given in eq. (C.1) for $(H_x/2\sqrt{2})^2 + (H_y/\sqrt{2})^2 \le 1$.

In the Voigt notation [71],

$$\delta\sigma_i'(\vec{r}) = \left[ \phi_{i1}'\partial_y^2 + \phi_{i2}'\partial_x^2 - \phi_{i3}'\partial_x\partial_y \right]\Psi'(\vec{r}),$$
$$\text{where} \qquad \phi_{ij}' = \lim_{R_0 \to 2a_0} \phi_{ij} = \mathcal{C}(2a_0, K)\delta_{ij}. \tag{67}$$

So in large length scales the $\hat{\Phi}$ tensor gives a difference between the coarse-grained local stresses for an amorphous system and a disordered-crystal. For an amorphous system $\hat{\Phi}$ is an identity matrix whereas in near-crystalline systems $\hat{\Phi}$ also contains off-diagonal elements. This difference vanishes for a marginally jammed disordered crystal i.e., $R_0 \to 2a_0$.

## 6.5 Stress correlations in three dimensions

In this section, we derive the stress correlations in a three dimensional system induced by microscopic disorder. The method developed earlier for the displacement fields and the change in local stresses is still valid for the 3D fcc lattice with the only difference arising from the number of nearest neighbors and their arrangement in space. We can also write the displacement fields of the particles as, $\delta\tilde{r}^\alpha(\vec{k}) = G^\alpha(\vec{k})\delta\tilde{a}(\vec{k})$ [35]. Using this we can write down the expression for change in the local stress components in an fcc lattice in Fourier space as, $\delta\tilde{\sigma}_{\alpha\beta}(\vec{k}) = S_{\alpha\beta}(\vec{k})\delta\tilde{a}(\vec{k})$. The exact expressions for Green's functions for displacement fields and the change in local stress components due to both particle size disorder and force pinning in fcc lattice are detailed in the appendix A.1 and appendix A.2 respectively. At large length-scales, the Green's function in real space for change in local stress has the following radial

behavior,

$$S_{\alpha\beta}(\vec{r}) = \mathcal{F}\left[S_{\alpha\beta}(\vec{k})\right] \sim \frac{B_{\alpha\beta}(\theta,\phi)}{r^3}, \qquad \forall r \gg R_0. \tag{68}$$

Similar to two dimensional systems, we can write the correlation between different components of stress using the Green's functions defined above in $|\vec{k}| \to 0$ limit as

$$C_{\alpha\beta\mu\nu}(\theta,\phi) = \langle \delta a^2 \rangle \lim_{|\vec{k}|\to 0} S_{\alpha\beta}(\vec{k}) S_{\mu\nu}(-\vec{k}). \tag{69}$$

Here the preliminary theoretical results for the Green's functions for local stress and the stress correlations in Fourier space are given in the Appendix A.2 and all the distinct components of these correlations are plotted in Figure. 7.

## 7 Response to a point force

Subsequently, we examine the change in the local stress components within a crystalline system caused by finite quenched forces. We start by introducing finite external quenched forces, represented as $\vec{f}_a(\vec{r}_i)$, to each grain $i$ in the crystalline system. The sum of these forces satisfies the condition $\sum_i \vec{f}_a(\vec{r}_i) = 0$. To balance the forces on each grain in the system, particles shift from their original lattice positions. In the case of an ideal crystalline system with forces attached to particles, the force balance requirement for each grain $i$ can be expressed as follows:

$$\sum_j \left( f_{ij}^{(0)\mu} + \delta f_{ij}^{\mu} \right) = -(f_a)_i^{\mu}, \tag{70}$$

where $f_{ij}^{(0)}$ are the forces along the bond between particle $i$ and $j$ in the initial crystalline system whereas $\delta f_{ij}$ correspond to the change in the bond forces due to the external pinning. Here $\delta f_{ij}$ can be approximated using their first-order Taylor series expansion, resulting in the following linear expression:

$$\sum_j \sum_\nu C_{ij}^{\mu\nu} \delta r_{ij}^{\nu} = -(f_a)_i^{\mu}. \tag{71}$$

The above expression is similar to the linear equation for the displacement fields due to particle size disorder. By employing a similar approach as explained earlier for particle size disorder, we can obtain the expression for displacement fields resulting from pinned forces, which is presented as follows:

$$\delta \tilde{r}^{\mu}(\vec{k}) = -\sum_\nu \underbrace{\left(A^{-1}\right)^{\mu\nu}(\vec{k})}_{\tilde{\mathcal{G}}^{\mu\nu}(\vec{k})} \tilde{f}_a^{\nu}(\vec{k}), \tag{72}$$

whose inverse Fourier transform gives the displacement fields in real space due to force quench. The large lengthscale behavior of these displacements has been studied thoroughly [32]. We have given a brief description of these above expressions in the Appendix A.1. Using these displacement fields and the expression for the change in local stresses, one can write the components of change in local stresses in Fourier space as,

$$\delta \tilde{\sigma}_{\alpha\beta}(\vec{k}) = \sum_\mu \mathcal{S}_{\alpha\beta}^{\mu}(\vec{k}) \tilde{f}_a^{\mu}(\vec{k}), \tag{73}$$

where

$$\mathcal{S}_{\alpha\beta}^{\mu}(\vec{k}) = \sum_j \left[ \left( -1 + F_j(\vec{k}) \right) \left( \sum_\nu \Delta_j^{\alpha(0)} C_j^{\beta\nu} \tilde{\mathcal{G}}^{\mu\nu}(\vec{k}) + f_j^{\beta(0)} \tilde{\mathcal{G}}^{\mu\alpha}(\vec{k}) \right) \right]. \tag{74}$$

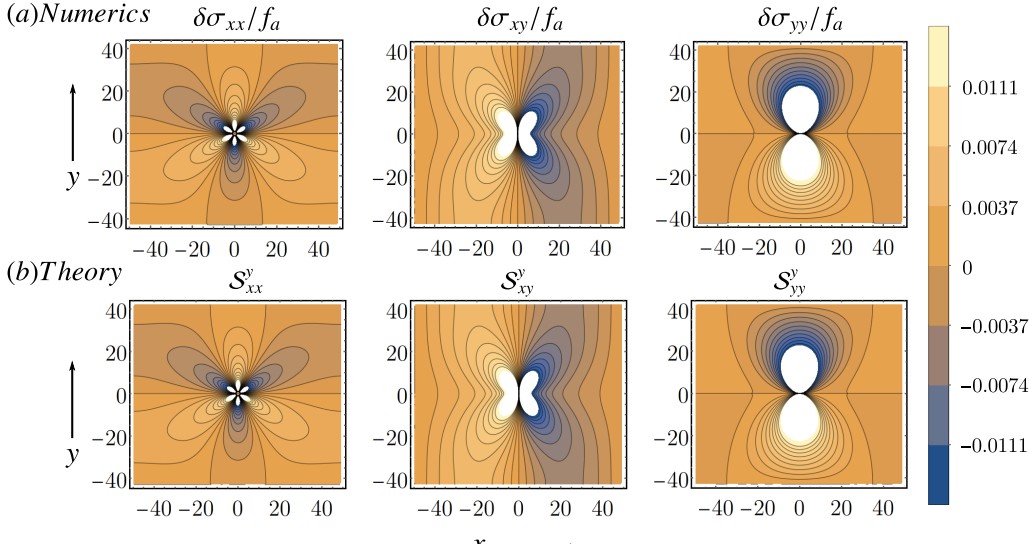

Figure 8: Change in local stress components per unit force ($\delta\sigma_{\alpha\beta}(\vec{r})/f_a$) applied at the origin in 2D near-crystalline systems of $N = 10000$ particles with periodic boundary conditions. Here the top panel ($a$) corresponds to direct numerical simulations and ($b$) represents the theoretical results as given in Eq. (78).

We can connect the Green's functions for the displacement and stress fields produced due to force quench to that of the microscopic particle size disorder which can be represented as follows,

$$
\begin{aligned}
\tilde{G}^\mu(\vec{k}) &= \sum_\nu \tilde{\mathcal{G}}^{\mu\nu}(\vec{k})\left(\sum_j C_j^{\nu a}\left[1 + F_j(\vec{k})\right]\right), \\
S_{\alpha\beta}(\vec{k}) &= \sum_\nu \mathcal{S}_{\alpha\beta}^\nu(\vec{k})\left(\sum_j C_j^{\nu a}\left[1 + F_j(\vec{k})\right]\right).
\end{aligned}
\tag{75}
$$

In real space, the change in the local stress components can be represented as

$$
\delta\sigma_{\alpha\beta}(\vec{r}) = \sum_{\vec{r}'}\sum_\mu \mathcal{S}_{\alpha\beta}^\mu(\vec{r}-\vec{r}')f_a^\mu(\vec{r}'),
\tag{76}
$$

where $\mathcal{S}_{\alpha\beta}^\mu(\vec{r})$ corresponds to the Green's function for the change in local stress components due to pinned force in real space. These Green's functions in the above expression can be understood as the change in local stresses due to a unit force applied to a single particle at the origin. The exact expressions for these Green's functions in Fourier space (in $k \to 0$ limit) are detailed in Appendix. A.2. We can perform Fourier transform to obtain the large lengthscale behavior of these Green's functions which shows $1/r$ radial behavior in two dimensions and $\mathcal{S}_{\alpha\beta}^\gamma \sim 1/r^2$ and $\mathcal{S}_{\alpha\alpha}^\alpha \sim 1/r$ in three dimensions. In two dimensions we can write these Green's functions explicitly as

$$
\begin{aligned}
\mathcal{S}_{xx}^x(\vec{r}) &= (4(R_0 - a_0)\cos\theta + a_0\cos 3\theta)/(2R_0 - a_0)r, \\
\mathcal{S}_{xy}^x(\vec{r}) &= (2R_0 - a_0 + 2a_0\cos 2\theta)\sin\theta/(2R_0 - a_0)r, \\
\mathcal{S}_{yy}^x(\vec{r}) &= a_0\cos 3\theta/(2R_0 - a_0)r, \\
\mathcal{S}_{xx}^y(\vec{r}) &= a_0\sin 3\theta/(2R_0 - a_0)r, \\
\mathcal{S}_{xy}^y(\vec{r}) &= (2R_0 - a_0 - 2a_0\cos 2\theta)\cos\theta/(2R_0 - a_0)r, \\
\mathcal{S}_{yy}^y(\vec{r}) &= (4(R_0 - a_0)\sin\theta - a_0\sin 3\theta)/(2R_0 - a_0)r.
\end{aligned}
\tag{77}
$$

To verify our results we have taken a system of $N = 10000$ particles in a triangular lattice arrangement and assigned a point force $(f_a\hat{y})$ directed along the $y-$axis to a particle located at the origin $(0,0)$. To make the net force in the system zero we apply an additional $-f_a\hat{y}/N$ to all the particles in the system. All the particles will rearrange to balance the point force, which results in a change in the local stress profile. So the displacements and the change in local stresses can be written as

$$\delta x(\vec{r}) = -f_a \mathcal{G}^{xy}(\vec{r}), \qquad \delta y(\vec{r}) = -f_a \mathcal{G}^{yy}(\vec{r}),$$
$$\delta\sigma_{\alpha\beta}(\vec{r}) = f_a \mathcal{S}^y_{\alpha\beta}(\vec{r}). \tag{78}$$

The results above can be verified in Fig. 8 where we have shown the match between the analytically and numerically obtained results for change in local stress profile due to a quenched force along $y$-axis. We have also done a preliminary study on the effect of pinning in 3d systems which we have detailed in the appendix A.2.

# 8 Distribution of stresses

Computations of stress correlations in athermal amorphous materials implicitly assume a partition function description [18, 43, 59, 60]. We show below that such an ensemble indeed emerges from the fluctuations of the individual radii. We obtain the joint probability distribution of changes in local stresses in Fourier space by utilizing the linear relationships between the change in local stress and disorder, as described in equations (40) and (76).

$$P(\delta\tilde{\sigma}_1(\vec{k}), \delta\tilde{\sigma}_2(\vec{k}), \delta\tilde{\sigma}_3(\vec{k})) = \int_{-\infty}^{\infty} d(\delta\tilde{a}_k) p(\delta\tilde{a}_k) \prod_{i=1}^{3} \delta\left(\delta\tilde{\sigma}_i^R - S_i \delta\tilde{a}(\vec{k})^R\right) \delta\left(\delta\tilde{\sigma}_i^I - S_i \delta\tilde{a}(\vec{k})^I\right)$$

$$= \int_{-\infty}^{\infty} \int_{-\infty}^{\infty} \prod_{j=1}^{3} df_j dg_j e^{i\left(f_j \delta\tilde{\sigma}_j^R + g_j \delta\tilde{\sigma}_j^I\right)} \underbrace{\int_{-\infty}^{\infty} d(\delta\tilde{a}_k) p(\delta\tilde{a}_k) e^{-i\left(\sum_{m=1}^{3} \left(\delta\tilde{a}(\vec{k})^R f_m + \delta\tilde{a}(\vec{k})^I g_m\right) S_m\right)}}_{h(\{f,g,S\})}. \tag{79}$$

In the above equation, the superscripts $R$ and $I$ correspond to the real and imaginary parts respectively. Since the $\delta as$ are drawn from a uniform distribution we can obtain the distribution of its Fourier transformed variable $\delta\tilde{a}(\vec{k})$ as $p(\delta\tilde{a}_k) = (48/\pi N\eta^2)^{1/2}\exp\{-48|\delta\tilde{a}(\vec{k})|^2/N\eta^2\}$. So using this distribution we can rewrite the above equation as,

$$h(\{f, g, S\}) = \exp\left\{-\frac{N\eta^2}{192}\sum_{m,n=1}^{3}(f_m f_n + g_m g_n)(S_m S_n)\right\} = \exp\left\{-\frac{N\eta^2}{192}(\langle f|\hat{S}|f\rangle + \langle g|\hat{S}|g\rangle)\right\}, \tag{80}$$

where $\langle f| = (f_1\ f_2\ f_3)$, $\langle g| = (g_1\ g_2\ g_3)$ and $\hat{S}_{mn} = S_m S_n$. Therefore, using the above expression we can rewrite the joint probability distribution of the stress components in Fourier space as

$$P(\delta\tilde{\sigma}_{xx}(\vec{k}), \delta\tilde{\sigma}_{yy}(\vec{k}), \delta\tilde{\sigma}_{xy}(\vec{k})) = \int_{-\infty}^{\infty}\int_{-\infty}^{\infty}\prod_{j=1}^{3} df_j dg_j e^{i\left(\langle f|\delta\tilde{\sigma}^R\rangle + \langle g|\delta\tilde{\sigma}^I\rangle\right)} e^{-\frac{N\eta^2}{192}\left(\langle f|\hat{S}|f\rangle + \langle g|\hat{S}|g\rangle\right)}$$

$$= \left(\frac{48}{\pi N\eta^2}\right)^{1/2} \exp\left\{-\frac{48}{N\eta^2}\langle\delta\tilde{\sigma}^R|\hat{S}^{-1}|\delta\tilde{\sigma}^R\rangle\right\}, \tag{81}$$

where

$$\left\langle \delta\tilde{\sigma}^R \right| = \begin{bmatrix} \delta\tilde{\sigma}_{xx}(\vec{k}) & \delta\tilde{\sigma}_{yy}(\vec{k}) & \delta\tilde{\sigma}_{xy}(\vec{k}) \end{bmatrix}, \quad \text{and} \quad \hat{S} = \begin{bmatrix} S_{xx}S_{xx} & S_{xx}S_{yy} & S_{xx}S_{xy} \\ S_{yy}S_{xx} & S_{yy}S_{yy} & S_{yy}S_{xy} \\ S_{xy}S_{xx} & S_{xy}S_{yy} & S_{xy}S_{xy} \end{bmatrix}, \quad (82)$$

where the form of the $S_{\alpha\beta}$ are model dependent and their exact forms are provided in the appendix A.2.

Earlier studies [18, 43] have shown that the generalized elastic constants in amorphous packings are directly related to the correlations in components of the stress tensor. This explicitly does not depend on the strength of the disorder in the system. However, as we have shown, in near-crystalline packings the distributions depend on the strength of the disorder but the elastic constants are independent of them. So it is not evident that the formulations developed earlier [18, 43] for generalised elastic constants in amorphous systems can be directly implemented in the context of near-crystalline packings.

## 9 Discussion and conclusion

In this study, we have examined the elastic properties, stress fluctuations as well as spatial stress correlations in near-crystalline athermal systems. We have obtained exact theoretical results for the macroscopic elastic properties by utilizing the fact that the average change in local stresses due to microscopic disorder in crystalline athermal solids is negligible. Our findings reveal that these elastic properties remain unaffected by the degree of disorder within a crystalline packing but are influenced by various initial conditions, such as packing fraction, pressure, and the strength of particle interactions. Furthermore, we have presented both numerical and theoretical results for local stress fluctuations and their spatial correlations within energy-minimized configurations of soft particles in both two and three dimensions. Notably, all these fluctuations and correlations exhibit a quadratic variation with the strength of the disorder. We have shown that the components of the stress tensor display anisotropic long range decay in both two and three dimensional near-crystalline packings irrespective of the interaction potential. For particle size disorder we observe a $1/r^d$ radial decay of the change in the stress-tensor components whereas for external quenched forces, we have established a slower $1/r^{d-1}$ radial decay. The stress correlations in disordered crystals differ significantly from those observed in isotropic amorphous materials at high packing fractions or under high-pressure conditions [59, 60]. Notably, we have observed additional non-isotropic angular behavior in the stress correlations, which becomes prominent at higher packing fractions.

Crucially, for the case of near-crystalline packings, we have found that the correlations depend on the strength of the disorder (proportional to $\eta^2$) introduced into the system, whereas the macroscopic elastic coefficients are largely independent of disorder. This is in contrast to stress correlations in amorphous materials, where the magnitude of the correlations have been related to elastic constants which are independent of the degree of disorder [18, 43]. It would therefore be very interesting to examine the crossover between this near-crystalline and amorphous behaviour as the degree of disorder increased beyond a critical threshold.

Several interesting questions still remain for further research. For example, the behavior of these stress correlations as we increase the disorder in the system may not vary quadratically to the strength of the disorder as linear perturbation expansion is not valid at a high enough disorder. It would therefore be interesting to study these properties across the crystalline to amorphous transition. It would also be intriguing to extend our analysis to dynamical systems where particles obey local force balance constraints only on average.

Recent studies [72] have connected the fluctuating elastic constants to the quasilocalised vibrational modes in amorphous materials. This hypothesis would be interesting to examine microscopically within the context of near-crystalline materials. Our study also highlights the importance of prestress in the stress correlations of athermal materials. Increasingly, studies have shown that accounting for the impact of stresses or prestress is crucial in understanding mechanical properties of amorphous materials [73,74]. Our techniques could therefore be used to test the impact of frozen-in stresses on the elasticity characteristics of both crystalline and amorphous materials.

# Acknowledgments

We thank Surajit Chakraborty, Pinaki Chaudhuri, Debankur Das, Bulbul Chakraborty, Subhro Bhattacharjee, Jishnu Nampoothiri and Palash Bera for useful discussions.

**Funding information**    The work of K. R. was partially supported by the SERB-MATRICS grant MTR/2022/000966. This project was funded by intramural funds at TIFR Hyderabad from the Department of Atomic Energy (DAE), Government of India.

# A Green's functions in the harmonic model

## A.1 Green's functions for displacement fields

The displacements of every grain (in Fourier space) due to pinned forces can be written using the translational invariance of the system as detailed in the recent publications [32,63] which have the form

$$\delta\tilde{r}^{\mu}(\vec{k}) = -\sum_{\nu}\underbrace{\left(A^{-1}\right)^{\mu\nu}(\vec{k})}_{\tilde{\mathcal{G}}^{\mu\nu}(\vec{k})}\tilde{f}_{a}^{\,\nu}(\vec{k}). \tag{A.1}$$

Similarly, the linear order displacement fields due to disorder in the particle sizes can be expressed as

$$\delta\tilde{r}^{\alpha}(\vec{k}) = \sum_{\nu}\tilde{G}^{\alpha}(\vec{k})\delta\tilde{a}^{\alpha}(\vec{k}). \tag{A.2}$$

Below we have given the general form of these Green's functions with various initial packing fractions, particle size disorder and random quenched forces for both two and three dimensional harmonic soft particle systems.



| Force pinning (2d) | | |
|---|---|---|
| $\tilde{\mathcal{G}}^{xx}(\vec{k})$ | $\frac{R_0 a_0^2}{K}\frac{m_1}{(m_1 m_2 - m_3^2)}$ | where, $m_1 = -3(R_0 - a_0) + (R_0 - 2a_0)\cos 2k_x + (2R_0 - a_0)\cos k_x \cos k_y\,,$ |
| $\tilde{\mathcal{G}}^{yy}(\vec{k})$ | $\frac{R_0 a_0^2}{K}\frac{m_2}{(m_1 m_2 - m_3^2)}$ | $m_2 = -3(R_0 - a_0) + R_0 \cos 2k_x + (2R_0 - 3a_0)\cos k_x \cos k_y\,,$ |
| $\tilde{\mathcal{G}}^{xy}(\vec{k})$ | $\frac{R_0 a_0^2}{K}\frac{m_3}{(m_1 m_2 - m_3^2)}$ | $m_3 = \sqrt{3}a_0 \sin k_x \sin k_y\,.$ |
| Force pinning (3d) | | |
| $\tilde{\mathcal{G}}^{\alpha\alpha}(\vec{k})$ | $\frac{R_0 a_0^2}{K}\frac{p_\beta p_\gamma - q_\alpha^2}{h}$ | where, $h = p_x p_y p_z + 2q_x q_y q_z - p_x q_x^2 - p_y q_y^2 - p_z q_z^2\,,$ |
| $\tilde{\mathcal{G}}^{\alpha\beta}(\vec{k})$ | $\frac{R_0 a_0^2}{K}\frac{p_\gamma q_\gamma - q_\alpha q_\beta}{h}$ | $p_\alpha = (-3R_0 + 4a_0 + (R_0 - 2a_0)\cos k_\beta \cos k_\gamma$ $+(R_0 - a_0)\cos k_\alpha(\cos k_\beta + \cos k_\gamma))\,,$ $q_\alpha = a_0 \sin k_\beta \sin k_\gamma\,, \qquad \forall\,\alpha \neq \beta \neq \gamma \in \{x, y, z\}\,.$ |
| Particle size disorder (2d) | | |
| $\tilde{G}^{\alpha}(\vec{k})$ | $\sum_\beta \tilde{\mathcal{G}}^{\alpha\beta}(\vec{k})D^\beta(\vec{k})$ | where, $D^x(\vec{k}) = i\frac{K(R_0 - a_0)}{a_0^3}\left(2\cos k_x + \cos k_y\right)\sin k_x\,,$ $D^y(\vec{k}) = i\frac{\sqrt{3}K(R_0 - a_0)}{a_0^3}\cos k_x \sin k_y\,.$ |
| Particle size disorder (3d) | | |
| $\tilde{G}^{\alpha}(\vec{k})$ | $\sum_\beta \tilde{\mathcal{G}}^{\alpha\beta}(\vec{k})D^\beta(\vec{k})$ | where, $D^\alpha(\vec{k}) = i\frac{K(R_0 - a_0)}{\sqrt{2}a_0^3}\left(\cos k_\beta + \cos k_\gamma\right)\sin k_\alpha,\ \forall\,\alpha \neq \beta \neq \gamma \in \{x, y, z\}\,.$ |

### A.2 Green's functions for change in local stresses in Fourier space

In the linear approximation, the change in the local stress due to external force pinning in both two and three dimensional systems can be expressed in Fourier space as

$$\delta\tilde{\sigma}_{\mu\nu}(\vec{k}) = \sum_\nu \mathcal{S}_{\mu\nu}^\alpha(\vec{k})\tilde{f}_a^\alpha(\vec{k})\,. \tag{A.3}$$

For particle size disorder the expressions for Green's functions $S_{\alpha\beta}$ as given in Eq. (39) which only depend on the nearest neighbor arrangement exhibit simple relationships at small magnitudes of $|\vec{k}|$ (corresponding to large lengthscales in real space), which are expressed below as

| Force Pinning (2d) | |
|---|---|
| $\mathcal{S}^\alpha_{\alpha\alpha}(\vec{k})$ | $-2ik_\alpha\left(k_\beta^2(4R_0-a_0)+k_\alpha^2(4R_0-5a_0)\right)/\left(k^4(2R_0-a_0)\right)$, |
| $\mathcal{S}^\alpha_{\beta\beta}(\vec{k})$ | $-2ik_\alpha(k_\alpha^2-3k_\beta^2)a_0/\left(k^4(2R_0-a_0)\right)$, |
| $\mathcal{S}^\alpha_{\alpha\beta}(\vec{k})=\mathcal{S}^\alpha_{\beta\alpha}(\vec{k})$ | $-2ik_\beta\left(k_\alpha^2(2R_0-5a_0)+k_\beta^2(2R_0-a_0)\right)/\left(k^4(2R_0-a_0)\right)$. |

| Force Pinning (3d) | |
|---|---|
| $\mathcal{S}^\alpha_{\mu\nu}(\vec{k})$ | $i\sqrt{2}R_0 n^\alpha_{\mu\nu}(\vec{k})/d(\vec{k})$ where, $$d(\vec{k})=c_0^2(c_0-a_0)k^6-\frac{3a_0^2}{2}c_0k^2\sum_{\alpha\beta\in\{x,y,z\}}k_\alpha^2k_\beta^2+5a_0^3(k_xk_yk_z)^2,$$ $$n^\alpha_{\alpha\alpha}(\vec{k})=c_0k^2\left[c_0^2k^2+2a_0c_1(k^2-k_\alpha^2)\right]-3a_0^2c_2k_\beta^2k_\gamma^2,$$ $$n^\alpha_{\beta\beta}(\vec{k})=a_0k_\alpha\left[2c_0c_1k^4+3a_0(8c_3-a_0)k_\beta^2(k_\alpha^2+k_\beta^2)+a_0c_0k^2k_\alpha^2\right.$$ $$\left.-(16c_3^2+a_0(2c_2+a_0))k^2k_\beta^2\right],$$ $$n^\alpha_{\alpha\beta}(\vec{k})=c_0k_\beta\left[2c_0c_3k^4-12R_0a_0k^2k_\alpha^2\right.$$ $$\left.-3a_0^2\left((4k_\alpha^2-k_\beta^2)(k_\alpha^2+k_\beta^2)-k^2(k_\alpha^2-k_\beta^2)\right)\right],$$ $$n^\alpha_{\beta\gamma}(\vec{k})=2a_0c_0k_\alpha k_\beta k_\gamma\left[2c_0k_\alpha^2+(2R_0+c_0)(k_\beta^2+k_\gamma^2)\right],$$ $\forall\,\alpha\neq\beta\neq\gamma\in\{x,y,z\}$. with $c_0=(2R_0-3a_0),c_1=(R_0-2a_0),c_2=(2R_0-a_0),c_3=(R_0-a_0)$. |

| Particle size disorder (2d) | |
|---|---|
| $S_{\alpha\alpha}(\vec{k})$ | $\frac{\mathcal{C}(R_0,K)}{|\vec{k}|^2}\left[\left(\frac{R_0}{a_0}-1\right)k_\beta^2+\left(2-\frac{R_0}{a_0}\right)k_\alpha^2\right]$, |
| $S_{\alpha\beta}(\vec{k})$ | $\frac{\mathcal{C}(R_0,K)}{|\vec{k}|^2}\left[\left(3-\frac{2R_0}{a_0}\right)k_\alpha k_\beta\right]$, $\qquad \forall\,\alpha\neq\beta\in\{x,y\}$, with $\mathcal{C}(R_0,K)=\frac{6KR_0(R_0-a_0)}{(2R_0-a_0)a_0^2}$. |

| Particle size disorder (3d) | |
|---|---|
| $S_{\mu\nu}(\vec{k})$ | $\left(2c_3R_0/a_0^3\right)\left(m_{\mu\nu}(\vec{k})/d(\vec{k})\right)$ where, $$d(\vec{k})=c_0^2(c_0-a_0)k^6-\frac{3a_0^2}{2}c_0k^2\sum_{\alpha\beta\in\{x,y,z\}}k_\alpha^2k_\beta^2+5a_0^3(k_xk_yk_z)^2,$$ $$m_{\alpha\alpha}(\vec{k})=c_0^3k^6-a_0^2(4c_1-a_0)k_\alpha^2k_\beta^2k_\gamma^2-c_0a_0k^2\left(4c_1k_\alpha^4+a_0^2k_\beta^2k_\gamma^2\right)$$ $$+c_0(8c_1^2+a_0c_0)k^4k_\alpha^2,$$ $$m_{\alpha\beta}(\vec{k})=c_0k_\alpha k_\beta\left(2c_0k^2-a_0(k_\alpha^2+k_\beta^2)\right)\left(2c_1k^2+a_0(k_\alpha^2+k_\beta^2)\right),$$ $\forall\,\alpha\neq\beta\neq\gamma\in\{x,y,z\}$. with $c_0=(2R_0-3a_0),c_1=(R_0-2a_0),c_2=(2R_0-a_0),c_3=(R_0-a_0)$. |

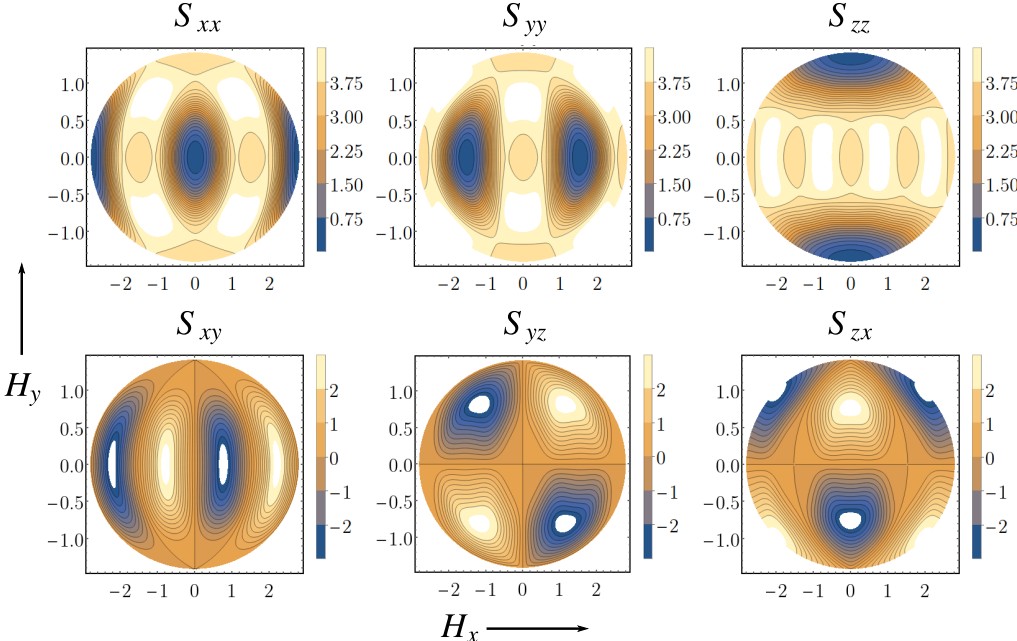

Figure 9: Green's function for change in the local stress components in $k \to 0$ limit which only has anisotropic angular behavior. Here the Green's functions are represented in Hammer projection i.e., $\{\theta, \phi \to H_x(\theta, \phi), H_y(\theta, \phi)\}$ as given in Eq. (C.1) for $(H_x/2\sqrt{2})^2 + (H_y/\sqrt{2})^2 \leq 1$.

In the 3D Harmonic model, when we approach the limit of $|\vec{k}| \to 0$ and $R_0 \to 2a_0$ with $a_0 = 1/2$, the components of $S_{\alpha\beta}$ take on specific forms:

$$S_{\alpha\alpha} = \frac{8(k_\beta^2 + k_\gamma^2)\left(|k|^4 - k_\beta^2 k_\gamma^2\right)}{|k|^6 + (k_x^6 + k_y^6 + k_z^6 + 2k_x^2 k_y^2 k_z^2)},$$

$$\text{and} \quad S_{\alpha\beta} = \frac{-8(k_\alpha k_\beta)\left(|k|^4 - k_\gamma^4\right)}{|k|^6 + (k_x^6 + k_y^6 + k_z^6 + 2k_x^2 k_y^2 k_z^2)}, \qquad \forall \alpha \neq \beta \neq \gamma \in \{x, y, z\}. \tag{A.4}$$

Now, in a spherical polar coordinate system, where $k_x = k \sin\theta \cos\phi$, $k_y = k \sin\theta \sin\phi$, and $k_z = k \cos\theta$, substituting these values into the equations above yields the angular behavior of $S_{\alpha\beta}(\vec{k})$ as $|\vec{k}| \to 0$,

$$\lim_{|\vec{k}| \to 0} S_{\alpha\beta}(|\vec{k}|, \theta, \phi) = \frac{g_{\alpha\beta}(\theta, \phi)}{h(\theta, \phi)}. \tag{A.5}$$

The angular behaviour of these Green's functions are represented in Fig. 9. In the specific case of $R_0 \to 2a_0$ with $a_0 = 1/2$, the components of $g_{\alpha\beta}(\theta, \phi)$ and $h(\theta, \phi)$ are given by,

$$\begin{aligned}
g_{xx} &= 8(\sin^2\theta \sin^2\phi + \cos^2\theta)\left(1 - \sin^2\theta \sin^2\phi \cos^2\theta\right), \\
g_{yy} &= 8(\sin^2\theta \cos^2\phi + \cos^2\theta)\left(1 - \sin^2\theta \cos^2\phi \cos^2\theta\right), \\
g_{zz} &= 8\sin^2\theta\left(1 - \sin^4\theta \sin^2\phi \cos^2\phi\right), \\
g_{xy} &= 2\sin^4\theta \sin^2 2\phi\left(1 - \cos^2\theta\right), \\
g_{yz} &= 2\sin^2 2\theta \sin^2\phi\left(1 - \sin^2\theta \cos^2\phi\right), \\
g_{zx} &= 2\sin^2 2\theta \cos^2\phi\left(1 - \sin^2\theta \sin^2\phi\right), \\
h(\theta, \phi) &= 1 + \sin^6\theta(\sin^6\phi + \cos^6\phi) + \cos^6\theta + 2\sin^4\theta \sin^2\phi \cos^2\phi.
\end{aligned} \tag{A.6}$$

# B   Angular variation of stress correlations at large lengthscales

All the six distinct stress correlations in a two dimensional Harmonic soft particle system where the underlying arrangement is a triangular lattice have the following forms,

$$
\begin{aligned}
C_{xxxx} &= 4\mathcal{P}(R_0, \eta)\left[\left(\frac{R_0}{a_0} - 1\right)\sin^2\theta + \left(2 - \frac{R_0}{a_0}\right)\cos^2\theta\right]^2 \xrightarrow{R_0 \to 2a_0} 4\mathcal{P}(2a_0, \eta)\sin^4\theta\,, \\
C_{yyyy} &= 4\mathcal{P}(R_0, \eta)\left[\left(\frac{R_0}{a_0} - 1\right)\cos^2\theta + \left(2 - \frac{R_0}{a_0}\right)\sin^2\theta\right]^2 \xrightarrow{R_0 \to 2a_0} 4\mathcal{P}(2a_0, \eta)\cos^4\theta\,, \\
C_{xyxy} &= 4\mathcal{P}(R_0, \eta)\left[\left(3 - \frac{2R_0}{a_0}\right)\sin\theta\cos\theta\right]^2 \xrightarrow{R_0 \to 2a_0} 4\mathcal{P}(2a_0, \eta)\sin^2\theta\cos^2\theta\,, \\
C_{yyxx} &= 4\mathcal{P}(R_0, \eta)\left[\sin^2\theta\cos^2\theta + \left(\frac{R_0}{a_0} - 1\right)\left(2 - \frac{R_0}{a_0}\right)(\sin^4\theta + \cos^4\theta)\right] \\
&\xrightarrow{R_0 \to 2a_0} 4\mathcal{P}(2a_0, \eta)\sin^2\theta\cos^2\theta\,, \\
C_{xxxy} &= 4\mathcal{P}(R_0, \eta)\left(3 - \frac{2R_0}{a_0}\right)\left[\left(\frac{R_0}{a_0} - 1\right)\sin^3\theta\cos\theta + \left(2 - \frac{R_0}{a_0}\right)\sin\theta\cos^3\theta\right] \\
&\xrightarrow{R_0 \to 2a_0} -4\mathcal{P}(2a_0, \eta)\sin^3\theta\cos\theta\,, \\
C_{yyyx} &= 4\mathcal{P}(R_0, \eta)\left(3 - \frac{2R_0}{a_0}\right)\left[\left(\frac{R_0}{a_0} - 1\right)\sin\theta\cos^3\theta + \left(2 - \frac{R_0}{a_0}\right)\sin^3\theta\cos\theta\right] \\
&\xrightarrow{R_0 \to 2a_0} -4\mathcal{P}(2a_0, \eta)\sin\theta\cos^3\theta\,.
\end{aligned}
\tag{B.1}
$$

The stress correlations shown above for marginally jammed crystals i.e., for $R_0 \to 2a_0$ have similar angular behavior as that of an isotropic amorphous material which have been shown earlier in several studies [18, 43, 58–60]. But for an overcompressed disordered crystal with finite average pressure, the angular behavior of stress correlations show an additional anisotropic term which depend on the system overcompression. Therefore for finite pressure, we can rewrite the above stress correlations of a disordered crystal as a summation of stress correlations of an isotropic amorphous material with the addition of a finite shift i.e., $C_{\alpha\beta\mu\nu} = C^t_{\alpha\beta\mu\nu} + d_{\alpha\beta\mu\nu}$, with

$$
\begin{aligned}
C^t_{xxxx} &= 4K_{2D}\sin^4\theta\,, & d_{xxxx}(\theta) &= 4K_{2D}t_0\left(t_0 + 2\sin^2\theta\right)\,, \\
C^t_{yyyy} &= 4K_{2D}\cos^4\theta\,, & d_{yyyy}(\theta) &= 4K_{2D}t_0\left(t_0 + 2\cos^2\theta\right)\,, \\
C^t_{xxxy} &= -4K_{2D}\sin^3\theta\cos\theta\,, & d_{xxxy}(\theta) &= -4K_{2D}t_0\sin(\theta)\cos(\theta)\,, \\
C^t_{yyxy} &= -4K_{2D}\sin\theta\cos^3\theta\,, & d_{yyxy}(\theta) &= 4K_{2D}t_0\sin(\theta)\cos(\theta)\,, \\
C^t_{xyxy} &= 4K_{2D}\sin^2\theta\cos^2\theta\,, & d_{xyxy}(\theta) &= 0\,, \\
C^t_{yyxx} &= 4K_{2D}\sin^2\theta\cos^2\theta\,, & d_{yyxx}(\theta) &= 4K_{2D}t_0(1 + t_0)\,,
\end{aligned}
\tag{B.2}
$$

where

$$
K_{2D} = \mathcal{P}(R_0, \eta)\left(\frac{2R_0}{a_0} - 3\right)^2\,, \quad \text{and} \quad t_0 = \frac{2a_0 - R_0}{2R_0 - 3a_0} \sim 2\left(1 - \left(\frac{\phi_0}{\phi}\right)^{1/2}\right)\,.
\tag{B.3}
$$

For an isotropic media with average pressure close to zero i.e $R_0 \to 2a_0$ both the theories for stress correlations dealing with two completely different scenarios give the same results as $d_{\alpha\beta\mu\nu} \to 0$.

## C  Hammer projection

The Hammer projection is a useful technique for visualizing functions that have a fixed radial coordinate, meaning they depend only on angular variables. This projection is particularly valuable in two dimensional visualizations. Any function with this characteristic can be effectively represented using Hammer projection. In Hammer projection, we transform from spherical coordinates $(\theta, \phi)$ to Hammer coordinates $(H_x, H_y)$ using the following equations:

$$H_x = \frac{2\sqrt{2}\cos(\theta - \pi/2)\sin(\phi/2)}{\sqrt{1 + \cos(\theta - \pi/2)\cos(\phi/2)}}, \qquad H_y = \frac{\sqrt{2}\sin(\theta - \pi/2)}{\sqrt{1 + \cos(\theta - \pi/2)\cos(\phi/2)}}. \qquad \text{(C.1)}$$

These equations allow us to map spherical coordinates to Hammer coordinates, facilitating the visualization of angular-dependent functions in a two dimensional space.

## D  Centrosymmetry order parmeter

The analytical expression for the local inversion symmetry order parameter can be represented as

$$F_{IS} = 1 - \frac{|\Xi|^2}{|\Xi|^2_{ISB}} = 1 - \frac{\sum_{i=1}^{N}\sum_{\alpha=x,y}\left(\sum_{jnni} n_{ij}^{\alpha} n_{ij}^{x} n_{ij}^{y}\right)^2}{\sum_{i,j}\left(n_{ij}^{x} n_{ij}^{y}\right)^2}, \qquad \text{(D.1)}$$

where $|\Xi|^2$ is the total affine force field of the disordered crystal for which $F_{IS}$ is being calculated and $|\Xi|^2_{ISB}$ is the force field of the asymmetric configuration where the inversion symmetry is completely broken. In Eq. (D.1), $n_{ij}^{\alpha}$ correspond to the $\alpha$ component of the unit vector joining two neighbouring particles $i$ and $j$. For any perfect crystal $F_{IS} = 1$.

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
