# Peer review of "Stress correlations in near-crystalline packings"

_SciPost Physics, doi:SciPost Phys. 17, 012 (2024)_

## Round 1 · Referee Report · Anonymous (Referee 1) · 2023-12-18

Strengths

1- detailed analysis of elasticity and stress correlations in nearly crystalline polydisperse solids 2- analytical results partly backed by numerical simulations 3- connection with other results in the literature

Weaknesses

1- some results about moduli are either unclear or in contradiction with previous results for similar systems 2- assumption of affine elasticity to be better justified

Report

I have read this manuscript "Stress correlations in near-crystalline packings" with much interest. The authors consider near-crystalline packings with finite polydispersity and compute the pressure tensor, elastic moduli and stress correlations both analytically and in simulations. The results are, overall, interesting and scientifically valid, however there are some problems and points, in the current version, that have to be solved and clarified before I can make a final recommendation.

  • the behaviour of the shear modulus in Fig. 2 b is puzzling. Why is the shear modulus decreasing with the volume fraction? This appears in contradiction with what is known for both disordered packings and crystals, where the shear modulus always increases with the packing fraction. This is because the affine shear modulus is proportional to the coordination number, which always increases monotonically with the packing fraction (or at least, it never decreases). Compare e.g. the analytical formulae of shear modulus (including both affine and nonaffine terms) in Phys. Rev. B 83, 184205 (2011) and for disordered crystals in Phys. Rev. B 93, 094204 (2016).

  • the assumption of affine elasticity in the analytical derivation should be better motivated. The root-cause of nonaffinity in the elasticity is the lack of inversion symmetry (centrosymmetry). If a particle is not a center of inversion symmetry, the forces coming from the nearest neighbours in the affine position do not balance, and the net force has to be released via the extra nonaffine displacements. The statistical degree of centrosymmetry can be easily evaluated using the order parameter defined in Phys. Rev. B 93, 094204 (2016) or the centrosymmetry command in LAMMPS https://docs.lammps.org/compute_centro_atom.html or by direct evaluation in the numerical simulations. Based on the above guidelines, the authors should provide a more quantitative justification as to why the nonaffine contributions to the moduli can be safely neglected. This is usually the case for the bulk modulus which is mostly affine even in disordered packings (https://www.nature.com/articles/srep18724) but typically this is not the case for the shear modulus.

  • in the context of Refs. 5,7,15 also Phys. Rev. B 83, 184205 (2011) should be mentioned.

  • anisotropic correlations in internal stresses (or elastic constants) have been used in Soft Matter, 2020,16, 7797-7807 within a rigorous field theory to predict the logarithmic Rayleigh scattering observed in Ref. 36. Without the anisotropic correlations in internal stress (or elastic constants), the logarithmic correction cannot be retrieved.

Requested changes

1- clarify or explain why shear modulus decreases with increasing packing fraction in Fig. 2b 2- quantify degree of centrosymmetry to justify assumption of affine elasticity in the derivation of the moduli 3- mention references and results listed in the report

  • validity: high
  • significance: high
  • originality: good
  • clarity: ok
  • formatting: excellent
  • grammar: excellent

Author:  Roshan Maharana  on 2024-01-22  [id 4268]

(in reply to Report 1 on 2023-12-18)
Category:
answer to question
reply to objection

We thank the Referee for the positive feedback on our manuscript. We have found the comments extremely useful and incorporated these points into the revised version of our manuscript. Below we address the points raised by the Referee in detail.

Comment 1: "The behaviour of the shear modulus in Fig. 2 b is puzzling. Why is the shear modulus decreasing with the volume fraction? This appears in contradiction with what is known for both disordered packings and crystals, where the shear modulus always increases with the packing fraction. This is because the affine shear modulus is proportional to the coordination number, which always increases monotonically with the packing fraction (or at least, it never decreases). Compare e.g. the analytical formulae of shear modulus (including both affine and nonaffine terms) in Phys. Rev. B 83, 184205 (2011) and for disordered crystals in Phys. Rev. B 93, 094204 (2016)."

Reply:

The referee is certainly correct about the fact that the affine shear modulus increases with increasing packing fraction in disordered solids as the coordination number also increases. But that is not the case in near crystalline systems where the coordination number remains fixed (i.e. $6$ in triangular lattice) with increasing packing fraction. As in near-crystalline systems, the average distance between two neighbouring particles is inversely proportional to square-root of the packing fraction (i.e. $R_0 \propto 1/\sqrt{\phi}$), with increasing packing fraction the average separation between particles decreases. Therefore for fixed shear amplitude, the particles are displaced less for higher packing fraction and the corresponding change in the shear stress is also smaller. This can be seen from the expression for the shear-stress for small shear amplitude which is given as, \begin{equation} \langle\Delta\Sigma_{xy}\rangle\sim -\gamma\frac{3Nk R_0}{2V a_0}\left(\frac{R_0}{2a_0}-\frac{3}{4}\right) = -\gamma\frac{\sqrt{3}k }{2a_0^2}\left(1-\frac{3}{4}\left(\frac{\phi}{\phi_0}\right)^{1/2}\right). \end{equation}

Changes in the manuscript: Based on the Referee's comment, we have included the citations to the analytical formulae for the shear modulus of disordered solids in the revised manuscript.

Comment 2: "the assumption of affine elasticity in the analytical derivation should be better motivated. The root cause of nonaffinity in the elasticity is the lack of inversion symmetry (centrosymmetry). If a particle is not a center of inversion symmetry, the forces coming from the nearest neighbours in the affine position do not balance, and the net force has to be released via the extra nonaffine displacements. The statistical degree of centrosymmetry can be easily evaluated using the order parameter defined in Phys. Rev. B 93, 094204 (2016) or the centrosymmetry command in LAMMPS https://docs.lammps.org/compute_centro_atom.html or by direct evaluation in the numerical simulations. Based on the above guidelines, the authors should provide a more quantitative justification as to why the nonaffine contributions to the moduli can be safely neglected. This is usually the case for the bulk modulus which is mostly affine even in disordered packings (https://www.nature.com/articles/srep18724) but typically this is not the case for the shear modulus.

  • in the context of Refs. 5,7,15 also Phys. Rev. B 83, 184205 (2011) should be mentioned."

Reply:

We thank the Referee for raising this important point. Indeed, there should be a better motivation for affine elasticity. In this study, we are calculating the elastic properties numerically with increasing magnitude of disorder. Additionally, we theoretically derive the elastic properties of the elastic properties for a perfect crystal without any disorder. Here the primary aim is to present a comparative analysis between the elastic properties of a crystal and that of a disordered crystal. It is important to note that we are not deducing the elastic properties of the disordered crystal, rather we derive them for the pure crystal and examine the threshold value of disorder where elastic properties begin to diverge.

Based on the Referee's comment, we have now evaluated the centrosymmetry order parameter in all near-crystalline arrangements, following the above references. This analysis indeed demonstrates the validity of the affine displacement assumption and provides a quantitative justification. We measure the average degree of centrosymmetry ($F_{IS}$) in the configurations which give, $F_{IS} \sim 1$ for $\eta \le 10^{-2}$. This indicates that for such near-crystalline systems, the non-affine contribution can be neglected.

Changes in the manuscript: Based on the Referee's comment, we have now evaluated the centrosymmetry order parameter and added a discussion regarding the motivation behind the affine displacement assumption in the manuscript. Additionally, we have included the references that were mentioned.

---

## Round 1 · Referee Report · Anonymous (Referee 1) · 2024-2-15

Report

The authors have done a very serious job in addressing all the referees' queries. The paper presents a valuable study of stress correlations in partly disordered packings. I recommend publication of the paper as is.
  • validity: high
  • significance: high
  • originality: high
  • clarity: good
  • formatting: excellent
  • grammar: excellent

Author:  Roshan Maharana  on 2024-04-08  [id 4394]

(in reply to Report 2 on 2024-02-15)

We thank the Referee for the positive feedback and recommendation of publication.

---

## Round 1 · Referee Report · Anonymous (Referee 2) · 2024-4-1

Strengths

1- detailed analytical analysis of stress correlations in near-crystalline solids

Report

In this manuscript, the authors introduce a new technique for calculating stress correlations in near-crystalline packings. The theoretical findings exhibit remarkable agreement with the simulation results, showcasing the efficacy of the developed theory. This advancement holds promise for investigating systems spanning both crystalline and amorphous solids. Given its potential impact, I believe this work is suitable for publication. I would recommend its publication after the authors address the following comments.

One main point highlighted by the authors is that the macroscopic elastic properties of near-crystalline packings remain unchanged below a given disorder threshold, which is demonstrated in Fig. 2(a). However, it will be beneficial to depict the deviation of or <G> from the theoretical value, B_theory/G_theory, of perfect crystals on a logarithmic scale, i.e., elucidating whether the deviations "B_theory - " and "G_theory - <G>" from nonaffine contribution also follow a power-law scaling with the strength of the disorder \eta. Such analysis could potentially influence the authors' conclusion that the macroscopic elastic coefficients are largely independent of the strength or the disorder.

Minor points:

  1. Some of the variables, such as \lambda_{SS}, \lambda_{SL}, and \lambda_{LL} in Eq. (4) and P_1 and P_2 in Eq. (23), are not defined when they are first introduced in the manuscript.

  2. Not all of the figures are referenced in the manuscript. In addition, the ordering of the figures in the manuscript does not match the order in which they are referenced.

  • validity: high
  • significance: good
  • originality: high
  • clarity: good
  • formatting: -
  • grammar: -

Author:  Roshan Maharana  on 2024-04-08  [id 4395]

(in reply to Report 3 on 2024-04-01)
Category:
answer to question
correction

We thank the Referee for the positive feedback on our manuscript. We provide detailed responses to the Referee below.

Comment 1: One main point highlighted by the authors is that the macroscopic elastic properties of near-crystalline packings remain unchanged below a given disorder threshold, which is demonstrated in Fig. 2(a). However, it will be beneficial to depict the deviation of $\langle B\rangle$ or $\langle G\rangle$ from the theoretical value, $B_{theory}/G_{theory}$, of perfect crystals on a logarithmic scale, i.e., elucidating whether the deviations "$B_{theory} - \langle B\rangle$" and "$G_{theory} - \langle G\rangle$" from nonaffine contribution also follow a power-law scaling with the strength of the disorder $\eta$. Such analysis could potentially influence the authors' conclusion that the macroscopic elastic coefficients are largely independent of the strength or the disorder.

Reply:

We thank the Referee for raising this important point. Based on a linear order perturbation expansion in disorder ($\eta$), our theory predicts that the the average modulus $\langle G\rangle$ and $\langle B\rangle$ remains independent of $\eta$ and are equal in value to those of a crystalline packing below a certain disorder magnitude ($\eta_c$), while the fluctuations in $G$ and $B$ will vary linearly with $\eta$. An earlier numerical work by Tong et. al., Scientific reports 5, 15378 (2015) has also found this behaviour. Our numerical results further suggest that for a small magnitude of disorder, both $B_{theory} - \langle B\rangle$ and $G_{theory} - \langle G\rangle$ do not have any power-law dependence on $\eta$.

In the attached figure titled "$Elastic\_moduli\_vs\_disorder.pdf$", we display the variation of the elastic moduli with increasing disorder strength. This shows a constant dependence of the moduli for a range of disorder strength. We also show the variation of the difference $G_{theory} - \langle G\rangle$ and $B_{theory} - \langle B\rangle$ in log scale, showing that the difference is negligible upto a certain threshold disorder value. This threshold value corresponds very closely with the transition point between crystals and disordered crystals. We have now incorporated these points and these plots in the revised version of the manuscript.

Comment 2: Some of the variables, such as $\lambda_{SS}, \lambda_{SL},$ and $\lambda_{LL}$ in Eq. (4) and $P_1$ and $P_2$ in Eq. (23), are not defined when they are first introduced in the manuscript.

Reply:

We thank the Referee for pointing this out.

Based on the Referee's comment, we have now defined all the above parameters in the revised manuscript.

Comment 3: Not all of the figures are referenced in the manuscript. In addition, the ordering of the figures in the manuscript does not match the order in which they are referenced.

Reply:

We thank the Referee for bringing this to our attention.

In the revised manuscript, we have ensured that all figures are referenced correctly and the ordering of the figures now aligns with the order in which they are referred.

Attachment:

Elastic_moduli_vs_disorder.pdf

---

## Round 2 · Author Response

List of changes
Below we write the list of changes in the revised manuscript.
-
Both Figure 1 and Figure 9 are cited in the main text.
-
The sequence of figures now corresponds to the order of citation.
-
Definitions for variables such as $\lambda_{SS},\lambda_{SL},\lambda_{LL}$ in Eq. (4) and $\mathcal{P}_1,\mathcal{P}_2$ in Eq. (23) are provided in sections 2.2 and 5, respectively.
-
Figure 2 has been updated to include the variation of the difference $B_{theory}-\langle B \rangle$ and $G_{theory}-\langle G \rangle$ in logarithmic scale.
-
Previous numerical finding regarding the variation of elastic modulus with disorder is cited on Page 7 as reference [31].
-
Centrosymmetry order parameter for near-crystalline packings are measured and written in Page 8.
-
Relevant citations for measuring the centrosymmetry order parameter are included as references [67], [68], and [69].
-
Appendix D now provides a comprehensive explanation of the measurement of Inversion symmetry order parameter.

---

## Round 2 · List of Changes

Below we write the list of changes in the revised manuscript.
-
Both Figure 1 and Figure 9 are cited in the main text.
-
The sequence of figures now corresponds to the order of citation.
-
Definitions for variables such as $\lambda_{SS},\lambda_{SL},\lambda_{LL}$ in Eq. (4) and $\mathcal{P}_1,\mathcal{P}_2$ in Eq. (23) are provided in sections 2.2 and 5, respectively.
-
Figure 2 has been updated to include the variation of the difference $B_{theory}-\langle B \rangle$ and $G_{theory}-\langle G \rangle$ in logarithmic scale.
-
Previous numerical finding regarding the variation of elastic modulus with disorder is cited on Page 7 as reference [31].
-
Centrosymmetry order parameter for near-crystalline packings are measured and written in Page 8.
-
Relevant citations for measuring the centrosymmetry order parameter are included as references [67], [68], and [69].
-
Appendix D now provides a comprehensive explanation of the measurement of Inversion symmetry order parameter.

---

## Editorial Decision

published